# Detecting Invariant Manifolds in ReLU-Based RNNs

## Abstract

Recurrent Neural Networks (RNNs) have found widespread applications in machine learning for time series prediction and dynamical systems reconstruction, and experienced a recent renaissance with improved training algorithms and architectural designs. Understanding why and how trained RNNs produce their behavior is important for scientific and medical applications, and explainable AI more generally. An RNN's dynamical repertoire depends on the topological and geometrical properties of its state space. Stable and unstable manifolds of periodic points play a particularly important role: They dissect a dynamical system's state space into different basins of attraction, and their intersections lead to chaotic dynamics with fractal geometry. Here we introduce a novel algorithm for detecting these manifolds, with a focus on piecewise-linear RNNs (PLRNNs) employing rectified linear units (ReLUs) as their activation function. We demonstrate how the algorithm can be used to trace the boundaries between different basins of attraction, and hence to characterize multistability, a computationally important property. We further show its utility in finding so-called homoclinic points, the intersections between stable and unstable manifolds, and thus establish the existence of chaos in PLRNNs. Finally we show for an empirical example, electrophysiological recordings from a cortical neuron, how insights into the underlying dynamics could be gained through our method.

## 1 Introduction

Recurrent neural networks (RNNs) are widely employed for time series forecasting [48, 79, 58, 30] and dynamical systems reconstruction [45, 70, 36, 6, 8, 65], especially in scientific applications like climate modeling [10, 59] or neuroscience [18], as well as in medical domains [44]. RNNs experienced a recent revival due to the advance of new powerful training algorithms that avoid vanishing or exploding gradients [36], and novel network architectures [70, 68, 62, 30, 69], in particular in the context of state space models (which are essentially linear RNNs with nonlinear readouts and input gating; [30, 56]). What lags behind, like in many other areas of deep learning, is a thorough theoretical understanding of the behavior of these systems and how they achieve the tasks they were trained on. Yet, such an understanding is crucial especially in scientific and medical areas where we are often interested in using trained RNNs as surrogate models for the underlying dynamical systems, providing mechanistic insight into the dynamical processes that generated the observed time series.

Formally, RNNs are recursive maps, hence discrete-time dynamical systems [52]. Dynamical systems theory (DST) offers a rich repertoire of mathematical tools for analyzing the behavior of such systems [64, 32]. Yet, the exploration of high-dimensional RNN dynamics remains a challenge due to limitations in current numerical methods, which struggle to scale and tend to yield only approximate results [76, 43, 29].

The dynamical behavior of a system is governed by its state space topology and geometry [2, 33], most prominently topological objects like attractors, such as stable fixed points, cycles, or chaotic sets, which determine the system's long-term behavior. Similarly important are the stable and unstable manifolds of fixed points and cycles, although they received much less attention in the scientific and ML communities. Stable manifolds are the sets of points in a system's state space that converge towards an equilibrium or periodic orbit in forward-time, while unstable manifolds, conversely, are the sets of points that converge to an equilibrium or periodic orbit in backward-time. Stable manifolds of saddle points delineate the boundaries between basins of attraction in multistable systems, that is, systems that harbor multiple attractor objects between which it can be driven back and forth by perturbations like external inputs or noise [22]. Multistability has been hypothesized to be an important computational property of RNNs, most prominently in computational neuroscience where it has been linked to working memory [17, 3, 9] or decision making [1, 81]. Different attractors may, for instance, correspond to different active memory states, and the process of convergence to one of these attractors to memory retrieval and pattern completion, as in Hopfield networks [38].

Tracing out stable and unstable manifolds is also important for finding homo- and heteroclinic orbits, which connect a cyclic point to itself or to another such point, respectively. These orbits provide a skeleton for the dynamics, forming structures like separatrix cycles [64] and heteroclinic channels [66, 39] which have been implied in flexible sequence generation. Intersections between stable and unstable manifolds also give rise to so-called homoclinic points which create sensitive regions in a system's state space associated with chaos [82]. Chaotic behavior, in turn, or regimes at the edge-of-chaos, have been associated with increased expressivity (larger function classes that can be emulated) and computational power in RNNs [16, 71, 5, 63, 49]. Chaos is also a practical problem in training RNNs, since it causes exploding gradients if not taken care of by special training techniques [52, 36].

Here we introduce a novel algorithm for computing the stable and unstable manifolds of fixed and cyclic points for the class of piecewise-linear (PL) RNNs (PLRNNs), which use piecewise nonlinearities like the rectified-linear unit (ReLU) as their activation function. Unlike traditional techniques designed for smooth dynamical systems—such as numerical continuation methods [47, 42] —the proposed method explicitly exploits the piecewise-linear structure of ReLU-based PLRNNs to enable the exact location of stable and unstable manifolds. In contrast to smooth systems, where numerical continuation techniques can be used to track invariant manifolds, methods for discrete-time systems are much more scarce, especially if these involve discontinuities in their Jacobians like ReLU-based RNNs. PLRNNs have emerged as one of the most powerful classes of models for dynamical systems reconstruction [36, 6, 8, 55], partly because the linear subspaces of such models support the indefinite online retention of memory contents without exploding or vanishing gradients [70, 56, 30, 31], and partly because their PL structure makes them more tractable [12]. For instance, efficient algorithms for exactly localizing fixed and cyclic points in polynomial time exist [19], on which we will partly build here. We illustrate how the algorithm can be employed to delineate the boundaries of basins of attraction and to detect homoclinic intersections, thus establishing the existence of chaos, and show how it can be used on empirical data to gain insight into dynamical mechanisms.

## 2 Related work

While there is an extensive literature by now on dynamical systems reconstruction with RNNs [6, 7, 36, 77, 60, 59, 65, 11, 69, 78], work on algorithms for dissecting topological properties of trained systems is much more scarce. Existing research is almost exclusively focused on algorithms for locating stable and unstable fixed points and cycles [29, 19], but not the invariant manifolds associated with them, despite their crucial role in structuring the state space and dynamics. More generally in the DST literature, a related class of methods are so-called continuation methods which numerically trace back important curves and manifolds in dynamical systems, mostly for systems of ordinary differential equations [47, 57]. Recently methods have been developed specifically for PL maps [74], but, similar to continuation methods, they seriously suffer from the curse of dimensionality. Thus, current methods effectively work only for very low-dimensional ($\leq 5d$) systems, in contrast to the size of many modern RNNs used for time series forecasting or dynamical systems reconstruction. The specific structure of PL maps, of which all types of PLRNNs are specific examples, considerably eases the computation of certain topological properties. This also provided the basis for the SCYFI algorithm for locating fixed points and cycles, which scales polynomially, in fact often linearly,

with the dimensionality of the PLRNN's latent space [19]. To our knowledge, no existing method efficiently detects stable and unstable manifolds in discrete-time RNNs. Our approach fills this gap by leveraging the piecewise-linear nature of ReLU-based RNNs to construct manifolds directly, bypassing the limitations of traditional techniques.

## 3 Methods

### 3.1 PLRNN architectures

A PLRNN [15] is simply a ReLU-based RNN with an additional linear term, which takes the basic form

$$\boldsymbol{z}_t = \boldsymbol{A}\boldsymbol{z}_{t-1} + \boldsymbol{W}\Phi(\boldsymbol{z}_{t-1}) + \boldsymbol{h}, \tag{1}$$

where $\boldsymbol{A} \in \mathbb{R}^{M \times M}$ is a diagonal matrix, $\boldsymbol{W} \in \mathbb{R}^{M \times M}$ an off-diagonal matrix, the element-wise ReLU non-linearity $\Phi(\cdot) = \max(0, \cdot)$, and $\boldsymbol{h} \in \mathbb{R}^M$ a bias term. Several variants of the PLRNN have been introduced to enhance its expressivity or reduce its dimensionality [6], of which we picked for demonstration here the shallow PLRNN (shPLRNN; [36]), $\boldsymbol{z}_t = \boldsymbol{A}\boldsymbol{z}_{t-1} + \boldsymbol{W}_1\Phi(\boldsymbol{W}_2\boldsymbol{z}_{t-1} + \boldsymbol{h}_2) + \boldsymbol{h}_1$ with $\boldsymbol{W}_1 \in \mathbb{R}^{M \times H}$, $\boldsymbol{W}_2 \in \mathbb{R}^{H \times M}$, and the recently proposed almost-linear RNN (ALRNN; [7]), which attempts to use as few nonlinearities as needed for the problem at hand, $\Phi(\boldsymbol{z}_t) = [\boldsymbol{z}_{1,t}, \dots, \boldsymbol{z}_{M-P,t}, \max(0, \boldsymbol{z}_{M-P+1,t}), \dots, \max(0, \boldsymbol{z}_{M,t})]$.

The piecewise linear structure of these PLRNNs can be exposed by rewriting the ReLU in Equation (1) as

$$\boldsymbol{z}_t = F_\theta(\boldsymbol{z}_{t-1}) = (\boldsymbol{A} + \boldsymbol{W}\boldsymbol{D}_{t-1})\boldsymbol{z}_{t-1} + \boldsymbol{h}, \tag{2}$$

$\boldsymbol{D_t} := \mathrm{diag}(\boldsymbol{d}_t)$ with $\boldsymbol{d}_t = (d_{1,t}, d_{2,t}, \dots, d_{M,t})$ and $d_{i,t} = 0$ if $z_{i,t} \leq 0$ and $d_{i,t} = 1$ otherwise [54]. For the ALRNN, we have $d_{1:M-P,t} = 1 \forall t$. Hence, we have $2^M$ linear subregions for the standard PLRNN, $2^H$ for the shPLRNN , and $2^P$ for the ALRNN.

### 3.2 Mathematical preliminaries

Recall that a fixed point of a recursive map $\boldsymbol{x}_t = F(\boldsymbol{x}_{t-1})$ is a point $\boldsymbol{x}^*$ for which $\boldsymbol{x}^* = F(\boldsymbol{x}^*)$, and a cyclic point with period $m$ is a fixed point of the $m$-times iterated map, i.e. such that $\boldsymbol{x}_{t+m} = F^m(\boldsymbol{x}_t) = \boldsymbol{x}_t$ (hence, a fixed point is a period-1 point). An $m$-cycle is a periodic sequence $\{\boldsymbol{x}_1^* \dots \boldsymbol{x}_m^*\}$ of such points with all points distinct, $\boldsymbol{x}_i^* \neq \boldsymbol{x}_j^* \; \forall 1 \leq i < j \leq m$.

**Definition 1** (**Un-/stable manifold**). Let $F : \mathbb{R}^M \to \mathbb{R}^M$ be a map and $\boldsymbol{p}$ be a hyperbolic period-$m$ cyclic point of $F$. The *local stable manifold* of $\boldsymbol{p}$, $W_{loc}^s(\boldsymbol{p})$, is defined as

$$W_{loc}^s(\boldsymbol{p}) := \{\boldsymbol{x} \in \mathbb{R}^M \; : \; F^{nm}(\boldsymbol{x}) \to \boldsymbol{p} \quad \text{as} \quad n \to \infty\}.$$

The *local unstable manifold* of $\boldsymbol{p}$, $W_{loc}^u(\boldsymbol{p})$, is defined as

$$W_{loc}^u(\boldsymbol{p}) := \{\boldsymbol{x} \in \mathbb{R}^M \; : \; F^{-nm}(\boldsymbol{x}) \to \boldsymbol{p} \quad \text{as} \quad n \to \infty\}.$$

The *global stable manifold* is the union of all preimages of the local stable manifold, and the *global unstable manifold* is created by the union of all images (forward iterations) of the local unstable manifold [61]:

$$W^s(\boldsymbol{p}) := \bigcup_{n=1}^{\infty} F^{-n}\Big(W_{loc}^s(\boldsymbol{p})\Big), \qquad W^u(\boldsymbol{p}) := \bigcup_{n=1}^{\infty} F^n\Big(W_{loc}^u(\boldsymbol{p})\Big). \tag{3}$$

If the map $F$ is noninvertible (i.e., does not have a unique inverse), the (global) stable manifold could be disconnected, making its computation hard as we need to trace back disconnected sets of points in time to determine it. However, most commonly we aim to approximate smooth continuous-time dynamical systems $\dot{\boldsymbol{x}} = f(\boldsymbol{x})$ in dynamical systems reconstruction through our RNN map $F_\theta$. For many important classes of such systems, like hyperbolic or Hamiltonian systems, the flow (solution) operator $\phi(t, \boldsymbol{x}_0) = \boldsymbol{x}_0 + \int_0^t f(\boldsymbol{u}) \, \mathrm{d}\boldsymbol{u}$ is often invertible, such that it is reasonable to assume (or enforce) invertibility for $F_\theta$ as well.

In each locally linear region of a PL dynamical system, the state space decomposes into invariant subspaces for the sets of positive and negative eigenvalues, respectively. For real eigenvalues,

trajectories lie on hyperplanes on which the motion is directed towards or away from the fixed point in a straight line. More generally, complex eigenvectors can occur and cause curvature, or eigenvalues may repeat, and the general dynamics in each affine subregion ($z_t = \tilde{W} z_{t-1} + h$, $\tilde{W} := A + W D(z_{t-1})$) is given by

$$z_t = \sum_{\substack{j=1 \\ j \neq i}}^{n} c_j \lambda_j^t v_j \; + \; \lambda_i^t \sum_{r=1}^{m} c_r \frac{t^{r-1}}{(r-1)!} w_r \; + \; \sum_{k=0}^{t-1} \tilde{W}^k h. \tag{4}$$

where the $c_i \in \mathbb{R}$ are determined from the initial condition, $v_i$ are the eigenvectors, $w_i$ are generalized eigenvectors (associated with eigenvalues with geometric multiplicity less than the algebraic, and for the multiplicity of $m > 1$). This formula describes the exponential evolution for all non-defective eigenvalues $\lambda_j$ and polynomially modulated exponentials for the defective eigenvalue $\lambda_i$, see Appx. C for details and special cases [64, 37].

Stable manifolds of saddle objects segregate the state space into different basins of attraction [24], which are sets of points from which the state evolves toward one or the other attractor. Formally, they are given by [2]:

**Definition 2** (**Basin of attraction**). Let $F : \mathbb{R}^M \to \mathbb{R}^M$ be a map. The *basin of attraction* of an attractor $\mathcal{A}$ is the largest open set $B(\mathcal{A}) \subseteq \mathbb{R}^M$ (containing $\mathcal{A}$) such that for every point $x \in B(\mathcal{A})$, the iterates of $x$ under the map $F$ converge to $\mathcal{A}$ in the forward limit:

$$B(\mathcal{A}) \; = \; \Big\{ x \in \mathbb{R}^M : d(F^k(x), \mathcal{A}) \to 0 \text{ as } k \to \infty \Big\},$$

where $F^k$ denotes the $k$-times iterated map $F$.

Stable and unstable manifolds are crucially important for the system dynamics not only because they dissect the state space into different regions of flow, but also because their intersections can give rise to complex types of dynamics like heteroclinic channels or chaos [66, 64, 75, 82].

**Definition 3** (**Homoclinic orbit**). Let $p$ be a saddle fixed point (or saddle cycle) of the map $F : \mathbb{R}^M \to \mathbb{R}^M$. A *homoclinic orbit* is a trajectory $\mathcal{O}_{hom} = \{x_n\}_{n \in \mathbb{N}}$ that connects $p$ to itself, i.e., $x_n \to p$ as $n \to \pm\infty$. Hence,

$$\mathcal{O}_{hom} \subset W^s(p) \; \cap \; W^u(p).$$

If $x \neq p$ is a point where the stable and unstable manifolds of $p$ intersect, then $x$ is referred to as a homoclinic point or intersection [61, 64]. Such an intersection of the stable and unstable manifolds leads to a horseshoe structure associated with a fractal geometry and chaos (see Appx. Def. 5, [82]).

**Definition 4** (**Heteroclinic orbit**). Let $p$ and $q$ be two *distinct* saddle fixed points (or saddle cycles) of the map $F$. A *heteroclinic orbit* $\mathcal{O}_{het} = \{x_n\}_{n \in \mathbb{N}}$ connects two different fixed points $p$ and $q$, that is, $x_n \to p$ as $n \to +\infty$ and $x_n \to q$ as $n \to -\infty$. Heteroclinic intersections (or points) can be defined similarly to homoclinic intersections, and, like homoclinic points, inevitably lead to chaos [82, 61].

### 3.3 Locating un-/stable manifolds

For a PL system, locally the unstable (resp. stable) manifolds of a cyclic point $p$ are simply given by the linear subspaces spanned by the eigenvectors with eigenvalues with positive (resp. negative) real part, and hence can easily be computed in closed form from the local Jacobians ($A + W D(p)$). However, as soon as we cross a border into a different linear subregion, the manifold may fold and start to follow different linear dynamics. For our algorithm, we can still exploit the fact that because the dynamics in the new subregion is again linear, the corresponding piece of the manifold will either be a (hyper-)plane segment or will have curvature along one or more directions according to Eq. 4, and will be of same dimensionality as the segment in the preceding linear subregion. To determine this segment, we only need a few support vectors to correctly position it in the new subregion. Assuming $F_\theta$ is invertible, we generate these by forward (unstable) or backward (stable) iterating the PLRNN map $F_\theta$. Invertibility also ensures a stable manifold continues along just one branch into a preceding subregion (instead of multiple disconnected sets).

Inverting Eq. 2 yields

$$z_{t-1} = (A + W D_{t-1})^{-1}(z_t - h). \tag{5}$$

**Algorithm 1** Manifold Construction

**Input:** $N_{max}$: Maximum number of iterations
$\quad\quad$ $P$ : periodic point
$\quad\quad$ $\sigma$: Index indicating whether the stable or unstable manifold is to be computed
$\quad\quad$ $E^\sigma$: Stable/unstable eigenvectors
$\quad\quad$ $\lambda^\sigma$: Stable/unstable eigenvalues

**Output:** $\{S_k^\sigma, Q_k^\sigma\}_{k=0}^{k_{max}}$: Sampled points and expression for the manifold in each subregion

1: $k=0$ $\quad\quad\quad\quad\quad\quad\quad\quad\quad\quad\quad\quad\quad\quad\quad\quad\quad\quad\quad\quad\quad\quad\quad\quad$ ▷ Index of linear subregion
2: $Q_k^\sigma \leftarrow$ GETMANIFOLD$(P, E^\sigma, \lambda^\sigma)$ $\quad\quad\quad\quad\quad\quad\quad\quad\quad$ ▷ Hyperplane or curved manifold
3: $S_k^\sigma \leftarrow$ SAMPLEPOINTS$(Q_k^\sigma)$ $\quad\quad\quad\quad\quad\quad\quad\quad\quad\quad$ ▷ Sample points on the manifold
4: **for** $n = 0 : N_{max}$ **do**
5: $\quad$ $S_1^\sigma \leftarrow$ PROPAGATETONEXTREGION$(S_k^\sigma)$ $\quad\quad\quad$ ▷ Propagate points to next subregion
6: $\quad$ $F^\sigma \leftarrow$ STEP$(S_1^\sigma)$ $\quad\quad\quad\quad\quad\quad\quad$ ▷ backward or forward step for flow vectors
7: $\quad$ $\{F^\sigma\}_{m=0}^M, \{S_1^\sigma\}_{m=0}^M \leftarrow$ UNIQUEREGIONS$()$ $\quad\quad\quad$ ▷ Sort points into unique subregions
8: $\quad$ $\{E^\sigma\}_{m=0}^M, \{\lambda^\sigma\}_{m=0}^M \leftarrow$ GETEIGEN$()$ $\quad\quad\quad$ ▷ Get eigenvectors/values in each region
9: $\quad$ **for** $m = 0 : M$ **do**
10: $\quad\quad$ $k+=1$
11: $\quad\quad$ $Q_{k+1}^\sigma \leftarrow$ GETMANIFOLD$(\{F^\sigma\}_m, \{S_1^\sigma\}_m, \{E^\sigma\}_m, \{\lambda^\sigma\}_m)$ $\quad\quad$ ▷ Linear or curved
12: $\quad\quad$ $S_{k+1}^\sigma \leftarrow$ SAMPLEPOINTS$(Q_{k+1}^\sigma)$ $\quad\quad\quad\quad\quad\quad$ ▷ Sample stable manifold
13: $\quad$ **end for**
14: **end for**
15: **return** $(\{S_k^\sigma\}_{k=0}^{k_{max}}, \{Q_k^\sigma\}_{k=0}^{k_{max}})$ $\quad\quad\quad\quad\quad\quad\quad$ ▷ Manifolds in every region

A solution to this eq. needs to be self-consistent, however, i.e. the signs of the $z_{i,t-1}$ on the l.h.s. need to be consistent with the entries $d_{i,t-1}$ in $\boldsymbol{D}_{t-1}$ on the r.h.s. To address this, we introduce a simple heuristic: 1. Perform a backward step using the current linear subregion $\boldsymbol{D}_t$. 2. Perform a forward step using the resulting candidate solution $\boldsymbol{z}_{t-1}^*$. 3. If $\boldsymbol{z}_t == F_\theta(\boldsymbol{z}_{t-1}^*)$, we are done. 4. If not, we take the candidate's linear subregion $\boldsymbol{D}(\boldsymbol{z}_{t-1}^*)$ to attempt a new inversion. 5. If this fails, we start checking the neighboring linear subregions by iteratively flipping bits in $\boldsymbol{D}_{t-1}$. See Algorithm 2 in the Appx. for details, with further subfunctions specified in Appx. H.2. Generally, this algorithm works for any ReLU based RNN. Its efficiency comes from the fact that usually the candidate will lie in the correct linear subregion, and – if this is not the case – it usually crosses only into neighboring regions.

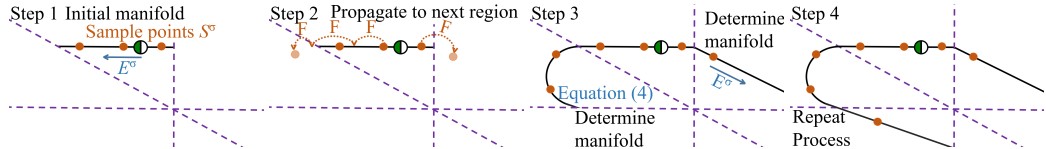

Figure 1: Illustration of the iterative procedure for computing stable manifolds with subregion boundaries indicated in purple-dashed. Step 1: The stable manifold (black) computation is initialized using the stable eigenvector (blue) of the saddle point (green), and sample points (orange) are placed along it. Step 2: These points are propagated until they enter another linear subregion, where the flow field is evaluated. Step 3: Using this, a new segment of the manifold is determined. Step 4: Repeating this process iteratively reconstructs the full global structure of the stable manifold. An example for a trained model can be found in Fig. 18.

We begin by using SCYFI [19] to identify all fixed and cyclic points. In the linear subregion which harbors the saddle point of interest, the stable or unstable manifold is spanned by the $n_{man}$ corresponding eigenvectors. To compute the global manifold, we sample seed points on this local manifold and propagate them backward or forward in time until a new linear subregion is reached, then take one additional step to capture the local flow. If the number of stable eigenvectors in the new subregion is the same as in the original subregion and the eigenvalues are non-degenerate, we can usually represent the hyperplane or the curved manifold using the eigenvectors (Eq. 4). Otherwise

we perform PCA (or kernel-PCA in the curved case), using the first $n_{man}$ components to span the manifold. We repeat this process, resampling and propagating, until all subregions have been visited or a desired depth is achieved (see Algo. 1 and Fig. 1). In systems with highly disparate timescales, this sampling process progresses fast along fast eigendirections and thereby tends to underrepresent slow eigendirections, causing numerical issues and poor approximations. To address this, we adjust sampling density inversely to eigenvalue magnitude. This balances contributions from different directions, improves accuracy for the slower directions, and ensures a more uniform representation.

### 3.4 Enforcing map invertibility by regularization

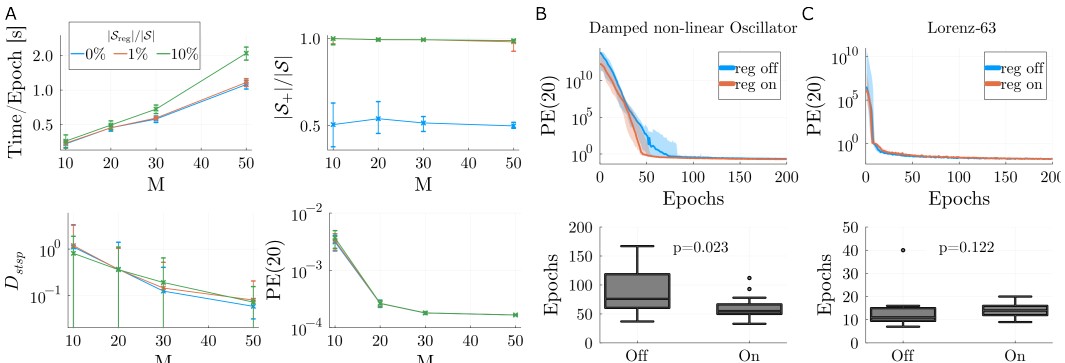

Figure 2: A) Runtime (top-left) and reconstruction quality (bottom) as a function of latent space dimensionality $M$ for different proportions $|\mathcal{S}_{reg}|/|\mathcal{S}|$ of subregions for which invertibility was enforced by regularization ($\lambda = 0.1 \exp(M)$ in Eq. 6). Means across 100 different training runs on the Lorenz-63 system $\pm$ SD are shown. Reconstruction quality was assessed through (dis-)agreement in attractor geometry (bottom-left; $D_{stsp}$, see Appx. J; [44, 6]) and 20-step-ahead prediction error (bottom-right). Top-right: Relative proportion of subregions with a positive determinant of the Jacobian ($\det(\boldsymbol{J}) > 0$), $\mathcal{S}_+$, as a function of latent space dimensionality $M$ for different proportions $|\mathcal{S}_{reg}|/|\mathcal{S}|$. Medians $\pm$ interquartile range are shown. B) Top: 20-step-ahead prediction error, PE(20), as a function of the number of training epochs when the invertibility regularization, Eq. 6, was turned off (blue) vs. on (orange), for a damped nonlinear oscillator. Bottom: Convergence to a predefined performance criterion (PE(20) $\leq 0.5$) was significantly faster (Mann-Whitney U-test) with the regularization turned on vs. off. Median across 20 trained models, error bands = interquartile range. C) Same as B for Lorenz-63 system.

In designing our algorithm, we relied on invertibility of the RNN map $F_\theta$, which is a reasonable assumption as the flow map $\phi(t, \boldsymbol{x}_0)$ for many important underlying ODE systems (incl. hyperbolic and Hamiltonian systems), which we attempt to approximate, is usually invertible. However, empirically, invertibility of $F_\theta$ is not always guaranteed (depending on the quality of the approximation). Thus, we enforce this condition by regularization.

$F_\theta$ is invertible, if the determinants of the Jacobian matrices of neighboring subregions have the same sign (sign condition) [23]. This can be enforced (for any type of ReLU-based RNN) by adding the following regularization term to the loss:

$$\mathcal{L}_{reg} = \lambda \cdot \frac{1}{|\mathcal{S}_{reg}|} \sum_{i \in \mathcal{S}_{reg}} \max\left(0, -\det(\boldsymbol{J}_i)\right), \tag{6}$$

computed across a small subset $\mathcal{S}_{reg}$ of linear subregions, where $\boldsymbol{J}_i = \boldsymbol{A} + \boldsymbol{W}\boldsymbol{D}_i$ is the Jacobian in subregion $i$, and $\lambda$ a regularization parameter. As shown in Fig. 2A, strategic sampling of only 1% of linear subregions, which are traversed by actual trajectories (cf. [7]), hardly affects runtime (Fig. 2A, top) and reconstruction performance (Fig. 2A, bottom) while still ensuring almost full invertibility (Fig. 2A, right). Since invertible flows are an inherent property of many, if not most, dynamical systems of scientific interest, we would furthermore expect that this regularization does not hamper, or even improves, the reconstruction of dynamical systems from data, in particular if the systems carry an intrinsic time reversibility [40]. This is confirmed in Fig. 2B-C which shows that with the invertibility regularization in place, training of a PLRNN on a 10d damped nonlinear oscillator converges significantly faster to a good solution ($p = 0.023$, Mann-Whitney U-test) than

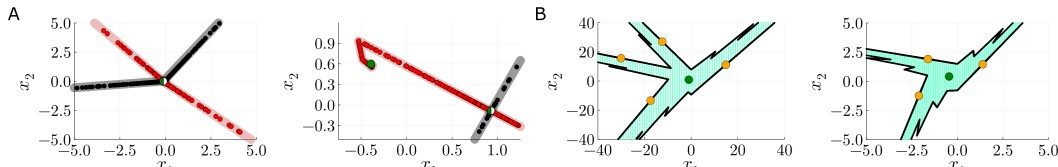

Figure 3: Model validation. A) Two examples of saddle points (half-green) with stable (gray solid lines) and unstable (red solid lines) manifolds determined by our algorithm, and points (black/ red dots, respectively) sampled by the analytical resp. backward/ forward map, showing that these all fall onto the analytically determined manifolds. B) Basins of attraction (light green, confirmed by sampling initial conditions and tracing their trajectories) of a stable fixed point (green dot) delineated by the stable manifold (black) of a 4-cycle (left) or 3-cycle (right).

without the regularization (Fig. 2B), while hardly affecting performance on other systems like the chaotic Lorenz-63 [50] system (Fig. 2C; $p = 0.122$).

## 4  Delineating basins of attraction

**Basic methods validation**  We first validate our algorithm on a simple toy example, a two-dimensional PL map for which we have analytical forms for the inverse and the fixed points (see Appx. D.1 for details). We chose parameters to produce a simple test case with only a single saddle point. Fig. 3A confirms that points produced by backward (resp. forward) iterating the map all lie on the stable (resp. unstable) manifold as determined by our algorithm. In this simple 2d example, the manifolds correspond to line segments. Changing the PL map's parameters slightly, we obtain a stable fixed point coexisting with a saddle period-4 (Fig. 3B, left) or period-3 (Fig. 3B, right) cyclic point [25]. Tracing back the stable manifold of the period-4 or period-3 saddle using our algorithm, we obtain the boundaries of the basins of attraction of the resp. fixed point (Fig. 3B). Fig. 3B further confirms this solution for the basin perfectly agrees with the 'classical' (not scalable) numerical approach of drawing initial conditions on a grid in state space, and observing their behavior in the limit $t \to \infty$.

**Bistable Duffing system**  The Duffing system [14] is a simple 2d nonlinear oscillator that can exhibit bistability between two spiral point attractors in certain parameter regimes (see Appx. I.1 for more details on the model). We trained a shPLRNN ($M = 2$, $H = 10$; [36]) on this system using sparse teacher forcing [52], and used SCYFI [19] to determine the fixed points. The basin boundary between the two spiral point attractors is the stable manifold of a saddle node in the center (Fig. 4A, and – as computed by our algorithm – agrees with the trajectory flows of the *true* system in blue.

**Multistable choice paradigm**  Simple models of decision making in the brain assume multistability between several choice-specific attractor states, to which the system's state is driven as one or the other choice materializes [80]. We trained an ALRNN ($M = 15$, $P = 6$) [7] to perform a simple 2-choice decision making task taken from [26], and, as before, use SCYFI to find fixed points and algorithm 1 to determine the stable manifold of a saddle separating the two basins of attraction corresponding to the two choices. The basin boundary consists of different linear pieces (hyperplanes) and is visualized in Fig. 4B in a 3d subspace (of the 15d system) together with the reconstructed system's fixed points (green) and some trajectories of the *true* system in blue.

**Lorenz-63 attractor**  The Lorenz-63 [50] model of atmospheric convection is probably the most famous example of a chaotic attractor. We reconstruct this system with a shPLRNN ($M = 3$, $H = 20$). Besides the chaotic attractor and two unstable spiral points, the system has a saddle node for which we compute the stable and unstable manifolds here. Fig. 4C confirms that these manifolds, as computed by algorithm 1 for the shPLRNN (left), agree well with those determined by numerical integration of the *original* Lorenz ODE system (right). This example also proves that the shPLRNN faithfully captured the geometrical structure of the state space, beyond just reconstruction of the chaotic attractor geometry itself.

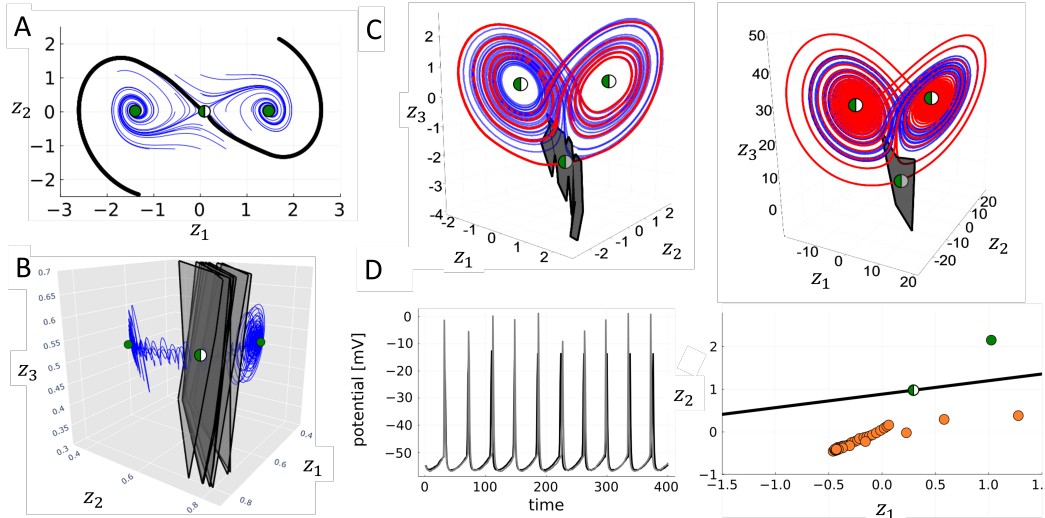

Figure 4: A) Reconstruction of the Duffing system by a shPLRNN ($M = 2$, $H = 10$). Trajectories drawn from the actual Duffing system in blue, identified fixed points in green, and in gray the stable manifold of the saddle in the center separating the two basins of attraction as determined by our algorithm. B) 3d subspace of the state space of an ALRNN ($M = 15$, $P = 6$) trained on a 2-choice decision making task, with true trajectories in blue. Two point attractors (green) were identified, with the stable manifold (black/gray) of a saddle (half-green) in the center separating the two basins. The stable manifold (basin boundary) consists of different planar pieces – note that for visualization these are projected down from a truly 15d space into a 3d subspace (accounting for some of the 'folded' appearance). C) Reconstruction for a shPLRNN ($M = 3$, $H = 20$) trained on the Lorenz63 system, with true trajectories in blue. The Lorenz63 system has two saddle-spirals in the center of the two lobes and a saddle at the bottom, which were correctly located by the shPLRNN. In black and red are the stable and unstable manifolds, respectively, of the saddle as identified by our algorithm (left), while on the right as computed by numerical continuation of the *original* Lorenz-63 system. The close agreement indicates the shPLRNN has correctly recovered the state space structure of the underlying system, although having been trained on trajectories from the actual attractor only. D) ALRNN ($M = 25$, $P = 6$) trained on electrophysiological recordings from a cortical neuron. Left: Time series of membrane voltage (true: gray, model-simulated: black); right: 2d projection of the ALRNN's state space with stable manifold of a saddle (black) separating the basins of attraction of a stable fixed point (green) and the 38-cycle (orange dots) corresponding to the spiking process. Note that the true stable manifold is a 24d curved object, which for visualization purposes is represented here by a locally linear approximation in the shown 2d subspace.

**Empirical example: Single cell recordings** In Figure 4D we trained an ALRNN ($M = 25$, $P = 6$) on membrane potential recordings from a cortical neuron [35]. The trained ALRNN contains a 38-cycle which corresponds to the rhythmic spiking activity in the real cell. In addition it has a stable fixed point and a saddle whose stable manifold (determined by our algorithm) separates the stable cycle from the stable fixed point, as illustrated in Fig. 4D (right), where we visualized the manifold through a locally linear approximation. Although we can compute the full global 24-dimensional manifold, its inherent curvature in the 25 dimensional state space makes it impossible to visualize it directly. Many types of cortical cells exhibit this type of bistability between spiking activity and a stable equilibrium near the resting or a more depolarized potential [41, 16, 13], and this example demonstrates how our algorithm can be utilized to reveal the structure of the state space supporting this type of dynamics from real cells.

## 5   Homo-/heteroclinic orbits and detection of chaos

The stable and unstable manifolds can be used to identify homoclinic and heteroclinic orbits, as defined in Sect. 3.2 (Def. 3, 4). Intersections between the stable and unstable manifolds of a saddle $p$ lead to homoclinic points, or to heteroclinic points of two saddles $p \neq q$. The existence of such

points inevitably gives rise to a complex fractal geometry (a so-scalled horseshoe structure, see Def. 5) and thus chaos [82]. Finding such intersections is therefore highly illuminating for determining the dynamical behavior of a system. This is illustrated for a simple 2d PL map in Fig. 5A, where homoclinic intersections were identified by algorithm 1. For this 2d case, we can in fact analytically determine the presence of homoclinic points, as worked out in Appx. H.3, the results of which agree with algorithm 1. Fig. 5B illustrates the resulting chaotic attractor (which lies on the unstable manifold), Fig. 5C the bifurcation diagram as one varies the model's bias term as a control parameter (showing the transition into the chaotic regime as $h_1$ falls below approximately 0.282), and Fig. 5D the system's two Lyapunov exponents across the chaotic range of $h_1$ (confirming the presence of 'robust chaos', as the Lyapunov exponents do not change across the chaotic regime [4]).

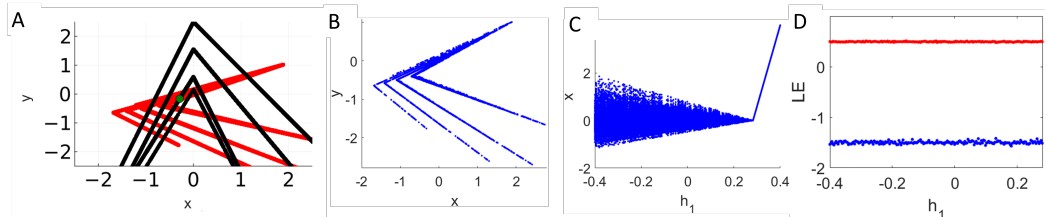

Figure 5: A) Stable (black) and unstable (red) manifolds of a saddle point (green dot) as identified by our algorithm. Note the homoclinic intersections between these two manifolds directly form the chaotic attractor. B) Structure of the chaotic attractor caused by these homoclinic intersections. C) Bifurcation diagram as a function of bias parameter $h_1$. D) Lyapunov exponents across the $h_1$-range for which the chaotic attractor exists.

# 6   Conclusions

Here we presented a novel semi-analytical algorithm for determining stable and unstable manifolds of fixed and cyclic points in ReLU-based RNNs. The algorithm analytically determines the local un-/stable manifolds within the subregion in which the cyclic point resides, and then by forward- or backward-iterating the RNN map $F_\theta$ collects a few support points for spanning their extensions into neighboring linear subregions. These manifolds are profoundly important for studying an RNN's dynamical repertoire, illuminating dynamical mechanisms in an underlying system reconstructed by the RNN, or the mechanisms by which an RNN solves a given task. Stable manifolds of saddle points segregate the state space into different basins of attraction, giving rise to the computationally important property of multistability [17, 22, 21]. Intersections of stable and unstable manifolds, in turn, lead to homo- or heteroclinic points which produce a fractal geometry and chaos [82, 61].

**Limitations**   One limitation of the proposed approach lies in its reliance on invertibility of the RNN map $F_\theta$. While this is a reasonable assumption for many systems of scientific interest, it nevertheless limits its applicability to certain classes of systems. It could still be applied to systems with non-invertible flows, but certain paths within the stable manifold may be inaccessible when tracing trajectories backward in time across linear subregions which differ in the signs of their Jacobian determinants. Enforcing invertibility also adds some computational burden to the training process, although a rather modest one if restricted to the set of subregions where most of the action happens (cf. Fig. 2A). Another limitation arises in the context of chaotic dynamics, where invariant manifolds may fold into fractal structures. While points from these manifolds may still be sampled, the analytic construction through curved/planar segments spanned by support vectors breaks down, as the intricate, self-similar geometry of fractals cannot be captured this way. Nevertheless, we still may be able to retrieve some important dynamical characteristics by determining homo- or heteroclinic intersections, as discussed in Section 5. Finally, in the worst case scenario the algorithm may scale as $2^P$ with the number of linear subregions $P$. However, as shown in [7], the number of subregions utilized by trained PLRNNs quickly saturates, suggesting an at most polynomial scaling if one restricts attention to the domain explored by the data (see also Fig.19). In any case, we emphasize that this is the *first* algorithm for detecting un-/stable manifolds in ReLU-based RNNs.

Code for the algorithms developed here and results produced is available at [upon publication].

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

# NeurIPS Paper Checklist

1. **Claims**

   Question: Do the main claims made in the abstract and introduction accurately reflect the paper's contributions and scope?

   Answer: [Yes]

   Justification: The results presented in the text and figures effectively support the claims made.

   Guidelines:

   - The answer NA means that the abstract and introduction do not include the claims made in the paper.
   - The abstract and/or introduction should clearly state the claims made, including the contributions made in the paper and important assumptions and limitations. A No or NA answer to this question will not be perceived well by the reviewers.
   - The claims made should match theoretical and experimental results, and reflect how much the results can be expected to generalize to other settings.
   - It is fine to include aspirational goals as motivation as long as it is clear that these goals are not attained by the paper.

2. **Limitations**

   Question: Does the paper discuss the limitations of the work performed by the authors?

   Answer: [Yes]

   Justification: Limitations are addressed in the limitations paragraph of the conclusion.

   Guidelines:

   - The answer NA means that the paper has no limitation while the answer No means that the paper has limitations, but those are not discussed in the paper.
   - The authors are encouraged to create a separate "Limitations" section in their paper.
   - The paper should point out any strong assumptions and how robust the results are to violations of these assumptions (e.g., independence assumptions, noiseless settings, model well-specification, asymptotic approximations only holding locally). The authors should reflect on how these assumptions might be violated in practice and what the implications would be.
   - The authors should reflect on the scope of the claims made, e.g., if the approach was only tested on a few datasets or with a few runs. In general, empirical results often depend on implicit assumptions, which should be articulated.
   - The authors should reflect on the factors that influence the performance of the approach. For example, a facial recognition algorithm may perform poorly when image resolution is low or images are taken in low lighting. Or a speech-to-text system might not be used reliably to provide closed captions for online lectures because it fails to handle technical jargon.
   - The authors should discuss the computational efficiency of the proposed algorithms and how they scale with dataset size.
   - If applicable, the authors should discuss possible limitations of their approach to address problems of privacy and fairness.
   - While the authors might fear that complete honesty about limitations might be used by reviewers as grounds for rejection, a worse outcome might be that reviewers discover limitations that aren't acknowledged in the paper. The authors should use their best judgment and recognize that individual actions in favor of transparency play an important role in developing norms that preserve the integrity of the community. Reviewers will be specifically instructed to not penalize honesty concerning limitations.

3. **Theory assumptions and proofs**

   Question: For each theoretical result, does the paper provide the full set of assumptions and a complete (and correct) proof?

Answer: [Yes]

Justification: The paper includes formal proofs in the Appendix.

Guidelines:

- The answer NA means that the paper does not include theoretical results.
- All the theorems, formulas, and proofs in the paper should be numbered and cross-referenced.
- All assumptions should be clearly stated or referenced in the statement of any theorems.
- The proofs can either appear in the main paper or the supplemental material, but if they appear in the supplemental material, the authors are encouraged to provide a short proof sketch to provide intuition.
- Inversely, any informal proof provided in the core of the paper should be complemented by formal proofs provided in appendix or supplemental material.
- Theorems and Lemmas that the proof relies upon should be properly referenced.

4. **Experimental result reproducibility**

Question: Does the paper fully disclose all the information needed to reproduce the main experimental results of the paper to the extent that it affects the main claims and/or conclusions of the paper (regardless of whether the code and data are provided or not)?

Answer: [Yes]

Justification: Pseudocode for all algorithms is provided in the paper, and together with the codebase, which will be released upon publication, all results will be reproducible.

Guidelines:

- The answer NA means that the paper does not include experiments.
- If the paper includes experiments, a No answer to this question will not be perceived well by the reviewers: Making the paper reproducible is important, regardless of whether the code and data are provided or not.
- If the contribution is a dataset and/or model, the authors should describe the steps taken to make their results reproducible or verifiable.
- Depending on the contribution, reproducibility can be accomplished in various ways. For example, if the contribution is a novel architecture, describing the architecture fully might suffice, or if the contribution is a specific model and empirical evaluation, it may be necessary to either make it possible for others to replicate the model with the same dataset, or provide access to the model. In general. releasing code and data is often one good way to accomplish this, but reproducibility can also be provided via detailed instructions for how to replicate the results, access to a hosted model (e.g., in the case of a large language model), releasing of a model checkpoint, or other means that are appropriate to the research performed.
- While NeurIPS does not require releasing code, the conference does require all submissions to provide some reasonable avenue for reproducibility, which may depend on the nature of the contribution. For example
  (a) If the contribution is primarily a new algorithm, the paper should make it clear how to reproduce that algorithm.
  (b) If the contribution is primarily a new model architecture, the paper should describe the architecture clearly and fully.
  (c) If the contribution is a new model (e.g., a large language model), then there should either be a way to access this model for reproducing the results or a way to reproduce the model (e.g., with an open-source dataset or instructions for how to construct the dataset).
  (d) We recognize that reproducibility may be tricky in some cases, in which case authors are welcome to describe the particular way they provide for reproducibility. In the case of closed-source models, it may be that access to the model is limited in some way (e.g., to registered users), but it should be possible for other researchers to have some path to reproducing or verifying the results.

5. **Open access to data and code**

Question: Does the paper provide open access to the data and code, with sufficient instructions to faithfully reproduce the main experimental results, as described in supplemental material?

Answer: [No]

Justification: The full codebase and data will be provided upon publication of the paper.

Guidelines:

- The answer NA means that paper does not include experiments requiring code.
- Please see the NeurIPS code and data submission guidelines (`https://nips.cc/public/guides/CodeSubmissionPolicy`) for more details.
- While we encourage the release of code and data, we understand that this might not be possible, so "No" is an acceptable answer. Papers cannot be rejected simply for not including code, unless this is central to the contribution (e.g., for a new open-source benchmark).
- The instructions should contain the exact command and environment needed to run to reproduce the results. See the NeurIPS code and data submission guidelines (`https://nips.cc/public/guides/CodeSubmissionPolicy`) for more details.
- The authors should provide instructions on data access and preparation, including how to access the raw data, preprocessed data, intermediate data, and generated data, etc.
- The authors should provide scripts to reproduce all experimental results for the new proposed method and baselines. If only a subset of experiments are reproducible, they should state which ones are omitted from the script and why.
- At submission time, to preserve anonymity, the authors should release anonymized versions (if applicable).
- Providing as much information as possible in supplemental material (appended to the paper) is recommended, but including URLs to data and code is permitted.

6. **Experimental setting/details**

Question: Does the paper specify all the training and test details (e.g., data splits, hyperparameters, how they were chosen, type of optimizer, etc.) necessary to understand the results?

Answer: [Yes]

Justification: The training details are provided in Appx. H.1

Guidelines:

- The answer NA means that the paper does not include experiments.
- The experimental setting should be presented in the core of the paper to a level of detail that is necessary to appreciate the results and make sense of them.
- The full details can be provided either with the code, in appendix, or as supplemental material.

7. **Experiment statistical significance**

Question: Does the paper report error bars suitably and correctly defined or other appropriate information about the statistical significance of the experiments?

Answer: [Yes]

Justification: Error bars are reported in figures with numerical results (Fig. 2)

Guidelines:

- The answer NA means that the paper does not include experiments.
- The authors should answer "Yes" if the results are accompanied by error bars, confidence intervals, or statistical significance tests, at least for the experiments that support the main claims of the paper.
- The factors of variability that the error bars are capturing should be clearly stated (for example, train/test split, initialization, random drawing of some parameter, or overall run with given experimental conditions).
- The method for calculating the error bars should be explained (closed form formula, call to a library function, bootstrap, etc.)

- The assumptions made should be given (e.g., Normally distributed errors).
- It should be clear whether the error bar is the standard deviation or the standard error of the mean.
- It is OK to report 1-sigma error bars, but one should state it. The authors should preferably report a 2-sigma error bar than state that they have a 96% CI, if the hypothesis of Normality of errors is not verified.
- For asymmetric distributions, the authors should be careful not to show in tables or figures symmetric error bars that would yield results that are out of range (e.g. negative error rates).
- If error bars are reported in tables or plots, The authors should explain in the text how they were calculated and reference the corresponding figures or tables in the text.

8. **Experiments compute resources**

Question: For each experiment, does the paper provide sufficient information on the computer resources (type of compute workers, memory, time of execution) needed to reproduce the experiments?

Answer: [Yes]

Justification: The details on computational resources are provided in Appx. H.1

Guidelines:

- The answer NA means that the paper does not include experiments.
- The paper should indicate the type of compute workers CPU or GPU, internal cluster, or cloud provider, including relevant memory and storage.
- The paper should provide the amount of compute required for each of the individual experimental runs as well as estimate the total compute.
- The paper should disclose whether the full research project required more compute than the experiments reported in the paper (e.g., preliminary or failed experiments that didn't make it into the paper).

9. **Code of ethics**

Question: Does the research conducted in the paper conform, in every respect, with the NeurIPS Code of Ethics https://neurips.cc/public/EthicsGuidelines?

Answer: [Yes]

Justification: No datasets with privacy concerns were used in this paper; the only one empirical dataset (physiological recordings from a neuron) will be open-sourced upon publication. The research is aimed at facilitating interpretability of machine learning models, making trained models explainable, and thus should primarily have positive ethical implications.

Guidelines:

- The answer NA means that the authors have not reviewed the NeurIPS Code of Ethics.
- If the authors answer No, they should explain the special circumstances that require a deviation from the Code of Ethics.
- The authors should make sure to preserve anonymity (e.g., if there is a special consideration due to laws or regulations in their jurisdiction).

10. **Broader impacts**

Question: Does the paper discuss both potential positive societal impacts and negative societal impacts of the work performed?

Answer: [NA]

Justification: This study represents basic research aimed at advancing analysis methods for trained models, thus helping to explain their behavior. We therefore anticipate no negative but, if anything, positive societal impact.

Guidelines:

- The answer NA means that there is no societal impact of the work performed.

- If the authors answer NA or No, they should explain why their work has no societal impact or why the paper does not address societal impact.
- Examples of negative societal impacts include potential malicious or unintended uses (e.g., disinformation, generating fake profiles, surveillance), fairness considerations (e.g., deployment of technologies that could make decisions that unfairly impact specific groups), privacy considerations, and security considerations.
- The conference expects that many papers will be foundational research and not tied to particular applications, let alone deployments. However, if there is a direct path to any negative applications, the authors should point it out. For example, it is legitimate to point out that an improvement in the quality of generative models could be used to generate deepfakes for disinformation. On the other hand, it is not needed to point out that a generic algorithm for optimizing neural networks could enable people to train models that generate Deepfakes faster.
- The authors should consider possible harms that could arise when the technology is being used as intended and functioning correctly, harms that could arise when the technology is being used as intended but gives incorrect results, and harms following from (intentional or unintentional) misuse of the technology.
- If there are negative societal impacts, the authors could also discuss possible mitigation strategies (e.g., gated release of models, providing defenses in addition to attacks, mechanisms for monitoring misuse, mechanisms to monitor how a system learns from feedback over time, improving the efficiency and accessibility of ML).

11. **Safeguards**

Question: Does the paper describe safeguards that have been put in place for responsible release of data or models that have a high risk for misuse (e.g., pretrained language models, image generators, or scraped datasets)?

Answer: [NA]

Justification: We do not anticipate a significant risk of misuse, see above.

Guidelines:

- The answer NA means that the paper poses no such risks.
- Released models that have a high risk for misuse or dual-use should be released with necessary safeguards to allow for controlled use of the model, for example by requiring that users adhere to usage guidelines or restrictions to access the model or implementing safety filters.
- Datasets that have been scraped from the Internet could pose safety risks. The authors should describe how they avoided releasing unsafe images.
- We recognize that providing effective safeguards is challenging, and many papers do not require this, but we encourage authors to take this into account and make a best faith effort.

12. **Licenses for existing assets**

Question: Are the creators or original owners of assets (e.g., code, data, models), used in the paper, properly credited and are the license and terms of use explicitly mentioned and properly respected?

Answer: [Yes]

Justification: All new code and results were created by us. Other packages were publically available and the respective authors and sources cited.

Guidelines:

- The answer NA means that the paper does not use existing assets.
- The authors should cite the original paper that produced the code package or dataset.
- The authors should state which version of the asset is used and, if possible, include a URL.
- The name of the license (e.g., CC-BY 4.0) should be included for each asset.
- For scraped data from a particular source (e.g., website), the copyright and terms of service of that source should be provided.

- If assets are released, the license, copyright information, and terms of use in the package should be provided. For popular datasets, `paperswithcode.com/datasets` has curated licenses for some datasets. Their licensing guide can help determine the license of a dataset.
- For existing datasets that are re-packaged, both the original license and the license of the derived asset (if it has changed) should be provided.
- If this information is not available online, the authors are encouraged to reach out to the asset's creators.

13. **New assets**

Question: Are new assets introduced in the paper well documented and is the documentation provided alongside the assets?

Answer: [Yes]

Justification: Pseudocode for all new algorithms is included in the paper and the full code will be made public upon publication.

Guidelines:

- The answer NA means that the paper does not release new assets.
- Researchers should communicate the details of the dataset/code/model as part of their submissions via structured templates. This includes details about training, license, limitations, etc.
- The paper should discuss whether and how consent was obtained from people whose asset is used.
- At submission time, remember to anonymize your assets (if applicable). You can either create an anonymized URL or include an anonymized zip file.

14. **Crowdsourcing and research with human subjects**

Question: For crowdsourcing experiments and research with human subjects, does the paper include the full text of instructions given to participants and screenshots, if applicable, as well as details about compensation (if any)?

Answer: [NA]

Justification: The paper did not involve crowdsourcing or research with human subjects.

Guidelines:

- The answer NA means that the paper does not involve crowdsourcing nor research with human subjects.
- Including this information in the supplemental material is fine, but if the main contribution of the paper involves human subjects, then as much detail as possible should be included in the main paper.
- According to the NeurIPS Code of Ethics, workers involved in data collection, curation, or other labor should be paid at least the minimum wage in the country of the data collector.

15. **Institutional review board (IRB) approvals or equivalent for research with human subjects**

Question: Does the paper describe potential risks incurred by study participants, whether such risks were disclosed to the subjects, and whether Institutional Review Board (IRB) approvals (or an equivalent approval/review based on the requirements of your country or institution) were obtained?

Answer: [NA]

Justification: The paper did not involved crowdsourcing or research with human subjects.

Guidelines:

- The answer NA means that the paper does not involve crowdsourcing nor research with human subjects.
- Depending on the country in which research is conducted, IRB approval (or equivalent) may be required for any human subjects research. If you obtained IRB approval, you should clearly state this in the paper.

- We recognize that the procedures for this may vary significantly between institutions and locations, and we expect authors to adhere to the NeurIPS Code of Ethics and the guidelines for their institution.
- For initial submissions, do not include any information that would break anonymity (if applicable), such as the institution conducting the review.

16. **Declaration of LLM usage**

Question: Does the paper describe the usage of LLMs if it is an important, original, or non-standard component of the core methods in this research? Note that if the LLM is used only for writing, editing, or formatting purposes and does not impact the core methodology, scientific rigorousness, or originality of the research, declaration is not required.

Answer: [NA]

Justification: The core method development does not involve LLMs. LLMs were exclusively used as support in editing and coding.

Guidelines:

- The answer NA means that the core method development in this research does not involve LLMs as any important, original, or non-standard components.
- Please refer to our LLM policy (`https://neurips.cc/Conferences/2025/LLM`) for what should or should not be described.

 **Appendix:**

## A    The PLRNN as 2d piecewise-linear map

The PLRNN is a piecewise linear map. As a simple example let us define a 2d map $F$ on $\mathbb{R}^2$ with one boundary, for which we have simple ground truth systems for, by:

$$F(X) = \begin{cases} \boldsymbol{A_l} \cdot X + \boldsymbol{B}, & \text{if } x \leq 0, \\ \boldsymbol{A_r} \cdot X + \boldsymbol{B}, & \text{if } x \geq 0, \end{cases} \tag{7}$$

where $X = \begin{pmatrix} x \\ y \end{pmatrix}$ is the position vector. The transformation matrices and vector are specified as follows:

$$\boldsymbol{A_l} = \begin{pmatrix} \tau_l & c \\ -\delta_l & d \end{pmatrix}, \quad \boldsymbol{A_r} = \begin{pmatrix} \tau_r & c \\ -\delta_r & d \end{pmatrix}, \quad \boldsymbol{B} = \begin{pmatrix} h_1 \\ h_2 \end{pmatrix}$$

This map can be reformulated as a PLRNN (Eq. 2) by defining the parameters $\boldsymbol{A}, \boldsymbol{W}, \boldsymbol{h}$ as:

$$\boldsymbol{A} = \begin{pmatrix} \tau_l & c \\ -\delta_l & d \end{pmatrix}, \quad \boldsymbol{W} = \begin{pmatrix} \tau_r - \tau_l & 0 \\ -\delta_r + \delta_l & 0 \end{pmatrix}, \quad \boldsymbol{h} = \begin{pmatrix} h_1 \\ h_2 \end{pmatrix}$$

## B    General 2d PL maps

A 2-unit PLRNN is a 2d PL dynamical system whose phase space is generally split into 4 sub-regions by 4 borders; see Appx. F for more details. For our exposition it suffices to focus on just one border, however. Hence studying general 2d PL maps, we can investigate dynamics of 2d PLRNNs locally near one border. Also, as shown in Appx. F, in some cases, there are only two different sub-regions divided by one border for 2d PLRNNs. This means in such situations, a 2d PLRNN exactly has the form of a 2d PL map. Therefore, here we examine dynamics of a 2d PL map.

A 2d PL map $T$ is a continuous map on $\mathbb{R}^2$ which is affine on each side of the line $\Sigma := \big\{ (x,y)^\mathsf{T} \in \mathbb{R}^2 : x = 0 \big\}$:

$$\begin{pmatrix} x^{(k+1)} \\ y^{(k+1)} \end{pmatrix} = T(x^{(k)}, y^{(k)})$$

$$= \begin{cases} T_\mathcal{L}(x^{(k)}, y^{(k)}) = \underbrace{\begin{pmatrix} a_l & c \\ b_l & d \end{pmatrix}}_{\boldsymbol{A_\mathcal{L}}} \begin{pmatrix} x^{(k)} \\ y^{(k)} \end{pmatrix} + \begin{pmatrix} h_1 \\ h_2 \end{pmatrix} ; \ \ x^{(k)} \leq 0 \\ \\ T_\mathcal{R}(x^{(k)}, y^{(k)}) = \underbrace{\begin{pmatrix} a_r & c \\ b_r & d \end{pmatrix}}_{\boldsymbol{A_\mathcal{R}}} \begin{pmatrix} x^{(k)} \\ y^{(k)} \end{pmatrix} + \begin{pmatrix} h_1 \\ h_2 \end{pmatrix} ; \ \ x^{(k)} \geq 0 \end{cases} \tag{8}$$

where $a_l, a_r, b_l, b_r, c, d, h_1, h_2 \in \mathbb{R}$. The phase space of the map Eq. 8 is divided into two different sub-regions $\mathcal{L} := \big\{ (x,y)^\mathsf{T} \in \mathbb{R}^2 : x \leq 0 \big\}$ and $\mathcal{R} := \big\{ (x,y)^\mathsf{T} \in \mathbb{R}^2 : x \geq 0 \big\}$ by the borderline $\Sigma$. The map Eq. 8 is continuous across the border, but its Jacobian matrix is discontinuous across $\Sigma$.

Under some conditions on the system parameters, there exists a well-defined and invertible coordinate change that transforms the general 2d PL map Eq. 8 into a 2d PL normal form map [28, 73]. For instance when $T$ is generic in the sense that $T(\Sigma)$ intersects $\Sigma$ at a unique point which is not a fixed point of $T$, or equivalently $c \neq 0$ and $(1 - d) h_1 + c h_2 \neq 0$; see [28] for more details. However, it is not always possible to apply such a transformation. Therefore, to study 2d PLRNNs in a comprehensive way, we need to investigate the map Eq. 8 in a general case, without any restrictions on the parameters, where here we focus particularly on the case $c = 0$ and/or $(1 - d) h_1 + c h_2 = 0$.

## C Stable manifolds with complex or degenerate eigenvalues

**2d spiral**  For complex-conjugate eigenvalues in 2d, the associated eigenvectors span a two-dimensional invariant plane in which the motion forms a spiral (or rotation with expansion/ contraction) around the equilibrium. That is, within each linear subregion, the solution on an un-/stable manifold evolves in a straight or spiraling fashion. The spiral structure can be derived from the eigenvalues as follows. For simplicity we shift the coordinate system such that $h = 0$. Suppose $\boldsymbol{A} + \boldsymbol{W}\boldsymbol{D}_t$ has a *complex* eigenvalue $\lambda = r\, e^{i\theta}$   (where $|\lambda| = r < 1$) with corresponding eigenvector $\boldsymbol{v} = \boldsymbol{A} + i\,\boldsymbol{b}, \quad \boldsymbol{A}, \boldsymbol{b} \in \mathbb{R}^d$. A solution in the direction of this eigenvector evolves as

$$\boldsymbol{z}_t \;=\; (\boldsymbol{A} + \boldsymbol{W}\boldsymbol{D}_t)^t\, \boldsymbol{z}_0.$$

If $\boldsymbol{z}_0$ lies in the span of $\boldsymbol{A}, \boldsymbol{b}$, then

$$\boldsymbol{z}_t \;=\; \mathrm{Re}\left(\lambda^t v\right) \;=\; r^t\Big[\cos(t\,\theta)\,\boldsymbol{A} \;-\; \sin(t\,\theta)\,\boldsymbol{b}\Big]. \tag{9}$$

**General case**

Consider a linear system of the form $\boldsymbol{z}_{t+1} = \boldsymbol{L}\boldsymbol{z}_t$.

**Case (1)**: If the geometric multiplicity equals the algebraic multiplicity, then $\boldsymbol{L}$ is diagonalizable and admits a basis of linearly independent eigenvectors $\boldsymbol{v}_1, \cdots, \boldsymbol{v}_n$. Thus, the orbits of the system can be expanded in terms of eigenvectors as

$$\boldsymbol{z}_t = c_1 \lambda_1^t \boldsymbol{v}_1 + c_2 \lambda_2^t \boldsymbol{v}_2 + \cdots + c_n \lambda_n^t \boldsymbol{v}_n. \tag{10}$$

**Case (2)**: If the geometric multiplicity of $\lambda_i$ is less than the algebraic multiplicity, then the eigenvalue is said to be defective and $\boldsymbol{L}$ admits a basis of generalized eigenvectors. In this case, $\boldsymbol{L}$ has a Jordan block $\boldsymbol{J}_m(\lambda_i)$ of size $m > 1$

$$\boldsymbol{J}_m(\lambda_i) \;=\; \begin{pmatrix} \lambda_i & 1 & 0 & \ldots & 0 \\ 0 & \lambda_i & 1 & \ldots & 0 \\ 0 & 0 & \lambda_i & \ldots & 0 \\ \vdots & \vdots & \vdots & \ddots & 1 \\ 0 & 0 & 0 & \ldots & \lambda_i \end{pmatrix},$$

with the $t$-th power

$$\boldsymbol{J}_m^t(\lambda_i) \;=\; \begin{pmatrix} \lambda_i^t & \binom{t}{1}\lambda_i^{t-1} & \binom{t}{2}\lambda_i^{t-2} & \cdots & \binom{t}{m-1}\lambda_i^{t-m+1} \\ 0 & \lambda_i^t & \binom{t}{1}\lambda_i^{t-1} & \cdots & \binom{t}{m-2}\lambda_i^{t-m+2} \\ \vdots & \vdots & \ddots & \ddots & \vdots \\ 0 & 0 & \ldots & \lambda_i^t & \binom{t}{1}\lambda_i^{t-1} \\ 0 & 0 & \ldots & 0 & \lambda_i^t \end{pmatrix}.$$

Moreover, a generalized eigenvector $\boldsymbol{w}_m \neq 0$ of degree $m$, corresponding to the defective eigenvalue $\lambda_i$, satisfies

$$(\boldsymbol{L} - \lambda\boldsymbol{I})^m \boldsymbol{w}_m = 0, \qquad \text{but } (\boldsymbol{L} - \lambda\boldsymbol{I})^{m-1}\boldsymbol{w}_m \neq 0, \tag{11}$$

and $\boldsymbol{L}$ has $m$ linearly independent generalized eigenvectors associated with $\lambda_i$. In fact, we can construct a chain of generalized eigenvectors $\{\boldsymbol{w}_1, \cdots, \boldsymbol{w}_m\}$ such that

$$(\boldsymbol{L} - \lambda\boldsymbol{I})\boldsymbol{w}_m = \boldsymbol{w}_{m-1}, \; (\boldsymbol{L} - \lambda\boldsymbol{I})\boldsymbol{w}_{m-1} = \boldsymbol{w}_{m-2}, \; \cdots, \; (\boldsymbol{L} - \lambda\boldsymbol{I})\boldsymbol{w}_2 = \boldsymbol{w}_1, \tag{12}$$

where $\boldsymbol{w}_1$ is a regular eigenvector. Given a chain of length $m$, the orbit contribution corresponding to the Jordan block $\boldsymbol{J}_m(\lambda_i)$ is given by

$$\boldsymbol{z}_t^{defective} \;=\; \lambda_i^t\Big(c_1\boldsymbol{w}_1 + c_2\,t\boldsymbol{w}_2 + c_3\,\frac{t^2}{2!}\boldsymbol{w}_3 + \cdots + c_m\,\frac{t^{m-1}}{(m-1)!}\boldsymbol{w}_m\Big). \tag{13}$$

The full orbit of the system is a linear combination of contributions from the non-defective eigenvalues $\lambda_j$, $1 \leq j \neq i \leq n$, and the defective eigenvalue $\lambda_i$ as

$$z_t = \sum_{\substack{j=1 \\ j \neq i}}^{n} c_j \lambda_j^t v_j + \lambda_i^t \sum_{r=1}^{m} c_r \frac{t^{r-1}}{(r-1)!} w_r. \tag{14}$$

This formula describes exponential evolution for all non-defective eigenvalues and polynomially modified exponentials for the defective eigenvalue $\lambda_i$.

For an affine system of the form $z_{t+1} = L z_t + h$, Eq. 14 consists of a homogeneous part (determined by the eigenstructure of $L$) and a particular part due to the constant bias term $h$. In the presence of a defective eigenvalue $\lambda_i$ with Jordan block of size $m$, the full orbit is given by

$$z_t = \sum_{\substack{j=1 \\ j \neq i}}^{n} c_j \lambda_j^t v_j \;+\; \lambda_i^t \sum_{r=1}^{m} c_r \frac{t^{r-1}}{(r-1)!} w_r \;+\; \sum_{k=0}^{t-1} L^k h. \tag{15}$$

The final term $\sum_{k=0}^{t-1} L^k h$ accounts for the cumulative effect of the bias, modifying the orbit away from purely exponential or polynomial-exponential behavior.

# D  Chaos

**Definition 5** (**Horseshoe structure**). A horseshoe structure is generated when there is a homoclinic/heteroclinic intersection between stable and unstable manifolds of a saddle fixed point and therefore an infinite number of similar intersections. The occurrence of a homoclinic/heteroclinic intersection implies the existence of a chaotic orbit [82].

**Definition 6** (**Robust chaos**). A chaotic attractor is called robust if, for its parameter values, there are no periodic windows and coexisting attractors in some neighborhood of the parameter space such that the chaotic attractor is unique in that open subset [4, 61]. Accordingly, small perturbations in either the parameter or phase space will not destroy a robust chaotic attractor.

Many practical applications of neural networks depend on such a robust chaotic mode for reliable operation [61]. Banerjee et al. [4] were the first to introduce the idea of robust chaos in the context of 2d PL normal form maps. They discussed the occurrence of robust chaos in 1d and 2d PWS systems, and derived the existence and stability conditions of robust chaos in 2d PL normal form maps. In [46], robust chaos and border collision bifurcations (BCBs) were studied for non-invertible 2d PL normal form maps. Later, in [27] the concept of robust chaos was revisited to provide a new set of tools for analyzing it. Afterwards, [28] illustrated a constructive approach to examine robust chaos, in the original parameter regime of [4], based on invariant manifolds and expanding cones. Very recently, Simpson [73] detected invariant expanding cones and presented a general method to identify robust chaotic attractors in 2d border-collision normal form maps. Here, we investigate the existence of chaos for the general PL system Eq. 8 where $T$ is an invertible map. For this purpose, we first discuss its invertibility.

## D.1  Invertibility of the map Eq. 8

The map Eq. 8 is invertible iff $D_{\mathcal{L}} D_{\mathcal{R}} > 0$ and

$$\begin{aligned}
x^{(k)} &= T^{-1}(x^{(k+1)}) \\
&= \begin{cases}
T_{\mathcal{L}}^{-1} = A_{\mathcal{L}}^{-1}(x^{(k+1)} - h); & \frac{\varphi^{\mathsf{T}}(x^{(k+1)} - h)}{D_{\mathcal{L}}} \leq 0 \\[2ex]
T_{\mathcal{R}}^{-1} = A_{\mathcal{R}}^{-1}(x^{(k+1)} - h); & \frac{\varphi^{\mathsf{T}}(x^{(k+1)} - h)}{D_{\mathcal{R}}} \geq 0
\end{cases}
\end{aligned}, \tag{16}$$

where $D_{\mathcal{L}/\mathcal{R}}$ is the determinant of $A_{\mathcal{L}/\mathcal{R}}$, and $\varphi^{\mathsf{T}} = e_1^{\mathsf{T}} adj(A_{\mathcal{L}}) = e_1^{\mathsf{T}} adj(A_{\mathcal{R}}) = (d \;\; -c)$, [72]. Therefore

$$\begin{pmatrix} x^{(k)} \\ y^{(k)} \end{pmatrix} = T^{-1}(x^{(k+1)}, y^{(k+1)})$$

$$
= \begin{cases}
\frac{1}{D_\mathcal{L}} \left[ \begin{pmatrix} d & -c \\ -b_l & a_l \end{pmatrix} \begin{pmatrix} x^{(k+1)} \\ y^{(k+1)} \end{pmatrix} + \begin{pmatrix} c\,h_2 - d\,h_1 \\ b_l\,h_1 - a_l\,h_2 \end{pmatrix} \right]; & \Phi \le 0 \\[20pt]
\frac{1}{D_\mathcal{R}} \left[ \begin{pmatrix} d & -c \\ -b_r & a_r \end{pmatrix} \begin{pmatrix} x^{(k+1)} \\ y^{(k+1)} \end{pmatrix} + \begin{pmatrix} c\,h_2 - d\,h_1 \\ b_r\,h_1 - a_r\,h_2 \end{pmatrix} \right]; & \Phi \ge 0
\end{cases}
\tag{17}
$$

where

$$
\Phi := \frac{\varphi^\mathsf{T}(\boldsymbol{x}^{(k+1)} - \boldsymbol{h})}{D_{\mathcal{L}/\mathcal{R}}}
$$

$$
= \frac{d}{D_{\mathcal{L}/\mathcal{R}}}(x^{(k+1)} - h_1) - \frac{c}{D_{\mathcal{L}/\mathcal{R}}}(y^{(k+1)} - h_2).
\tag{18}
$$

Denoting the border of $T$ by $\Sigma = \{(x, y) \in \mathbb{R}^2 : x = 0\}$, the switching border of $T^{-1}$ is the line

$$
\boldsymbol{\ell}^\Sigma =: T(\Sigma) = \{(x, y) \in \mathbb{R}^2 : c\,y = d\,x - d\,h_1 + c\,h_2\},
\tag{19}
$$

along which $T^{-1}$ is continuous.

*Remark* D.1. For $c \ne 0$, the switching border of $T^{-1}$ is the line

$$
\boldsymbol{\ell}^\Sigma = \{(x, y) \in \mathbb{R}^2 : y = \frac{d}{c}x - \frac{d\,h_1}{c} + h_2\},
\tag{20}
$$

and $T(\Sigma)$ intersects $\Sigma$ at a unique point. While for $c = 0$ and $d \ne 0$ we have

$$
\boldsymbol{\ell}^\Sigma = \{(x, y) \in \mathbb{R}^2 : x = h_1\},
\tag{21}
$$

which means $T(\Sigma)$ and $\Sigma$ never intersect at any point or are coinciding with each other.

When $D_\mathcal{L} D_\mathcal{R} \le 0$, then either there is no inverse for $T$ or there are two inverses. If $D_\mathcal{L} D_\mathcal{R} > 0$, then there is a unique inverse $T^{-1}$ for $T$. In this case, for $c\,d \ne 0$, the nature of $T^{-1}$ depends on the sign of $D_\mathcal{L}$, $D_\mathcal{R}$, $c$, $d$ and $y^{(k+1)} - \dfrac{d(x^{(k+1)} - h_1)}{c} - h_2$; see Table 1. As shown in Table 2 (left), for $c = 0$ the nature of the inverse map can be determined based on the sign of $D_\mathcal{L}$, $D_\mathcal{R}$, $d$ and $x^{(k+1)} - h_1$. Note that, when $c = 0$, for $a_{l/r} = 1$ or $d = 1$ the matrix $\boldsymbol{A}_{\mathcal{L}/\mathcal{R}}$ has an eigenvalue 1. Similarly, for $d = 0$, it depends on the sign of $D_\mathcal{L}$, $D_\mathcal{R}$, $c$ and $y^{(k+1)} - h_2$; Table 2 (right).

Next we examine the theoretical conditions for the existence of chaotic orbits. In this regard, the fold structure of the stable and unstable manifolds will be discussed in detail. In particular it will be shown that, unlike 2d PL normal form maps, for general 2d PL maps the unstable manifold does not fold along the $x$-axis.

## D.2 Nature of the stable and unstable manifolds

Let $T$ be invertible and $\mathcal{O}_{\mathcal{L}/\mathcal{R}}$ a saddle fixed point with stable and unstable eigenvalues $\lambda_s$ and $\lambda_u$. The stable and unstable subspaces $E^s$ and $E^u$ of $\mathcal{O}_{\mathcal{L}/\mathcal{R}}$ are lines crossing $\mathcal{O}_{\mathcal{L}/\mathcal{R}}$, generated by the stable and unstable eigenvectors $\boldsymbol{v}_s$ and $\boldsymbol{v}_u$ associated with $\lambda_s$ and $\lambda_u$. Since $T$ is a PL map, the *local* stable and unstable manifolds of $\mathcal{O}_{\mathcal{L}/\mathcal{R}}$ coincide with the stable and unstable subspaces $E^s$ and $E^u$, respectively. The global stable and unstable manifolds $W^s(\mathcal{O}_{\mathcal{L}/\mathcal{R}})$ and $W^u(\mathcal{O}_{\mathcal{L}/\mathcal{R}})$ have a complex PL structure due to the PL nature of $T$. In fact, as mentioned before, any point on the $y$-axis maps to the line $\boldsymbol{\ell}^\Sigma$ as defined in Eq. 19. Since the linear map changes across the $y$-axis, the unstable manifold must have different slopes on the two sides of $\boldsymbol{\ell}^\Sigma$. That is, the unstable manifold folds along the line $\boldsymbol{\ell}^\Sigma$ (Fig.s 6-7) and all images of the fold points will also be fold points. Likewise, since under the action of $T^{-1}$ the line $\boldsymbol{\ell}^\Sigma$ maps to the $y$-axis, the stable manifold folds along the $y$-axis; see Fig. 7. Moreover, all the preimages of fold points are fold points. Because of the folds of $W^s(\mathcal{O}_{\mathcal{L}/\mathcal{R}})$ and $W^u(\mathcal{O}_{\mathcal{L}/\mathcal{R}})$, strictly speaking, they are not manifolds. However, they are called manifolds just as a matter of convention.

| $D_\mathcal{L}$ | $D_\mathcal{R}$ | $c$ | $d$ | $y^{(k+1)} - \dfrac{d(x^{(k+1)} - h_1)}{c} - h_2$ | Map |
|---|---|---|---|---|---|
| +ve | +ve | +ve | +ve | −ve | $T_\mathcal{R}^{-1}$ |
| +ve | +ve | +ve | −ve | −ve | $T_\mathcal{R}^{-1}$ |
| +ve | +ve | −ve | −ve | +ve | $T_\mathcal{R}^{-1}$ |
| +ve | +ve | −ve | +ve | +ve | $T_\mathcal{R}^{-1}$ |
| −ve | −ve | +ve | +ve | +ve | $T_\mathcal{R}^{-1}$ |
| −ve | −ve | +ve | −ve | +ve | $T_\mathcal{R}^{-1}$ |
| −ve | −ve | −ve | −ve | −ve | $T_\mathcal{R}^{-1}$ |
| −ve | −ve | −ve | +ve | −ve | $T_\mathcal{R}^{-1}$ |
| +ve | +ve | +ve | +ve | +ve | $T_\mathcal{L}^{-1}$ |
| +ve | +ve | +ve | −ve | +ve | $T_\mathcal{L}^{-1}$ |
| +ve | +ve | −ve | −ve | −ve | $T_\mathcal{L}^{-1}$ |
| +ve | +ve | −ve | +ve | −ve | $T_\mathcal{L}^{-1}$ |
| −ve | −ve | +ve | +ve | −ve | $T_\mathcal{L}^{-1}$ |
| −ve | −ve | +ve | −ve | −ve | $T_\mathcal{L}^{-1}$ |
| −ve | −ve | −ve | −ve | +ve | $T_\mathcal{L}^{-1}$ |
| −ve | −ve | −ve | +ve | +ve | $T_\mathcal{L}^{-1}$ |

Table 1: Signs of $D_\mathcal{L}$, $D_\mathcal{R}$, $c$, $d$ and $y^{(k+1)} - \dfrac{d(x^{(k+1)} - h_1)}{c} - h_2$ decide which map to apply.

| $D_\mathcal{L}$ | $D_\mathcal{R}$ | $d$ | $x^{(k+1)} - h_1$ | Map | $D_\mathcal{L}$ | $D_\mathcal{R}$ | $c$ | $y^{(k+1)} - h_2$ | Map |
|---|---|---|---|---|---|---|---|---|---|
| +ve | +ve | +ve | +ve | $T_\mathcal{R}^{-1}$ | +ve | +ve | +ve | −ve | $T_\mathcal{R}^{-1}$ |
| +ve | +ve | −ve | −ve | $T_\mathcal{R}^{-1}$ | +ve | +ve | −ve | +ve | $T_\mathcal{R}^{-1}$ |
| −ve | −ve | +ve | −ve | $T_\mathcal{R}^{-1}$ | −ve | −ve | +ve | +ve | $T_\mathcal{R}^{-1}$ |
| −ve | −ve | −ve | +ve | $T_\mathcal{R}^{-1}$ | −ve | −ve | −ve | −ve | $T_\mathcal{R}^{-1}$ |
| +ve | +ve | +ve | −ve | $T_\mathcal{L}^{-1}$ | +ve | +ve | +ve | +ve | $T_\mathcal{L}^{-1}$ |
| +ve | +ve | −ve | +ve | $T_\mathcal{L}^{-1}$ | +ve | +ve | −ve | −ve | $T_\mathcal{L}^{-1}$ |
| −ve | −ve | −ve | −ve | $T_\mathcal{L}^{-1}$ | −ve | −ve | −ve | +ve | $T_\mathcal{L}^{-1}$ |
| −ve | −ve | +ve | +ve | $T_\mathcal{L}^{-1}$ | −ve | −ve | +ve | −ve | $T_\mathcal{L}^{-1}$ |

Table 2: Left ($c = 0$): signs of $D_\mathcal{L}$, $D_\mathcal{R}$, $d$ and $x^{(k+1)} - h_1$, and right ($d = 0$): signs of $D_\mathcal{L}$, $D_\mathcal{R}$, $c$ and $y^{(k+1)} - h_2$ to determine the nature of $T^{-1}$.

Consider system Eq. 8 with parameters

$$a_r = 1.5, \ b_r = -0.75, \ a_l = -1.77, \ b_l = -0.9, \ c = 0.6,$$
$$d = 0.15, \ h_2 = -0.4, \ h_1 = -0.7. \tag{22}$$

For these values, $D_\mathcal{L} D_\mathcal{R} = 0.18529 > 0$ which implies the map Eq. 8 is invertible and $\mathcal{O}_\mathcal{L}^* = (-0.28848, -0.16514)^\mathsf{T}$ is a saddle fixed point in the left sub-region. The matrix $\boldsymbol{A}_\mathcal{L}$ has two real eigenvalues $\lambda_1 = -1.4277$ and $\lambda_2 = -0.1922$. Moreover, $\ell_u^*$ hits the border $x = 0$ at $P_0 = (0, -0.000582)^\mathsf{T}$, and one obtains $P_1 = (-0.70035, -0.40009)^\mathsf{T} \in \mathcal{L}$, and $P_2 = (0.29957, 0.1703)^\mathsf{T} \in \mathcal{R}$. Since $\mathsf{L}(x_1, y_1) \cdot \mathsf{L}(x_2, y_2) = 0.044708 > 0$, we need to determine the first and second fold points of the stable manifold. Calculating $\tilde{P}_0 = (0, 0.59343)^\mathsf{T}$ and $\frac{\varphi^\mathsf{T}(\tilde{P}_0 - \boldsymbol{h})}{D_\mathcal{L}} = -1.7889 < 0$, we get $\tilde{P}_{-1} = (-1.7889, -4.1106)^\mathsf{T}$. Hence, we observe $\tilde{\mathsf{L}}(x_1, y_1) \cdot \tilde{\mathsf{L}}(x_2, y_2) = -1.0269 < 0$, which implies the unstable manifold intersects the stable manifold at $\tilde{P}_{hom} = (-0.28848, -0.16514)^\mathsf{T}$. Since $\tilde{x}_{home} \tilde{x}_{-1} = 0.51606 > 0$ this is a homoclinic intersection and thus there must be a chaotic orbit, see Fig. 8. Moreover, fixing all parameter values while varying $h_1$ from negative to positive, a border collision bifurcation happens at $h_1 \approx 0.282$ ( Fig. 8B). For the parameter setting Eq. 22 we have $(1 - d) h_1 + c h_2 = 0.85 h_1 - 0.24$.

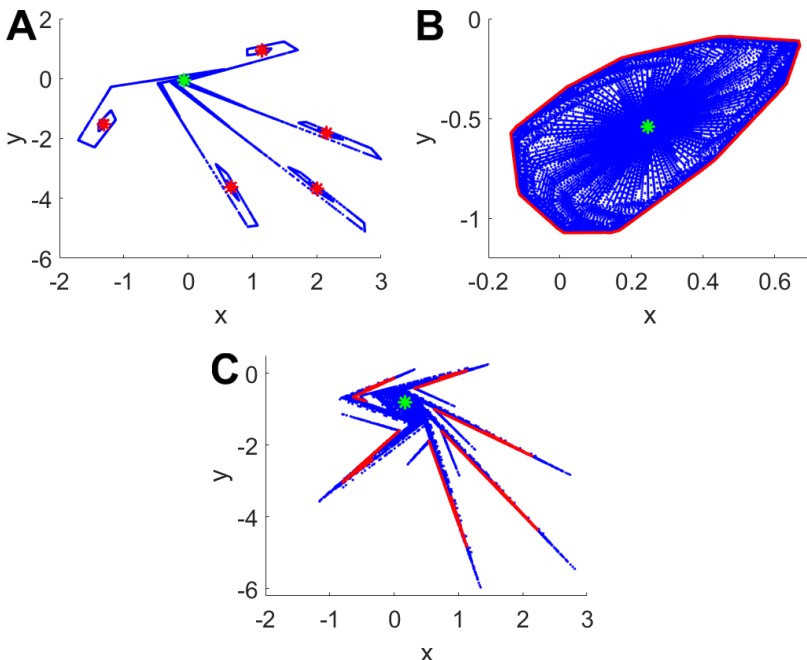

Figure 6: $A$) Unstable manifold (blue) corresponding to the saddle fixed point $\mathcal{O}_{\mathcal{L}}^* = (-0.0601, -0.0509)^{\mathsf{T}}$ (green star) and a stable 5-cycle (red stars) lying on it. Parameter settings: $a_r = 1.5$, $b_r = -1.58$, $a_l = -1.67$, $b_l = -0.9$, $c = 0.6$, $d = 0.1$, $h_2 = -0.1$, $h_1 = -0.13$. $B$) Unstable manifold corresponding to an unstable fixed point (green star) and a stable quasi-periodic orbit (red) on it. Parameter settings: $a_l = 0.31$, $b_l = -0.6$, $a_r = -1.1$, $b_r = -1.54$, , $c = 0.9$, $d = 0.3$, $h_1 = 1$, $h_2 = 0$. $C$) Unstable manifold of the unstable fixed point $\mathcal{O}_{\mathcal{R}}^* = (0.16978, -0.80815)^{\mathsf{T}}$ (green star) and a 2-band chaotic attractor (red) on it. Parameter settings: $a_r = 1.5$, $b_r = -1.69$, $a_l = -1.77$, $b_l = -0.9$, $c = 0.6$, $d = 0.15$, $h_2 = -0.4$, $h_1 = 0.4$.

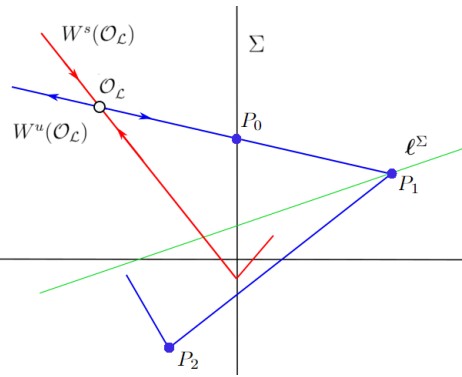

Figure 7: Folding structure of the stable and unstable manifolds of the map Eq. 8 for $c \neq 0$.

Therefore, at the bifurcation value $h_1 \approx 0.282$ the term $(1 - d)\, h_1 + c\, h_2$ becomes zero. Hence, this is a bifurcation value that cannot be obtained by considering the 2d PL normal form map.

In Fig. 8C, the Lyapunov exponents are plotted while parameter $h_1$ is varied. The largest (maximum) Lyapunov exponent (red) is positive throughout the displayed range, whereas the second one (blue) remains negative. This implies the orbit is chaotic throughout a larger range of parameter values, and hence the system exhibits *robust chaos* not sensitive to smaller changes in parameter values. With parameter values close to those in Eq. 22 such that $c = 0$, one still obtains chaotic attractors (Fig. 9). Recall that for $c = 0$ the general 2d PL map Eq. 8 cannot be transformed into a 2d PL normal form

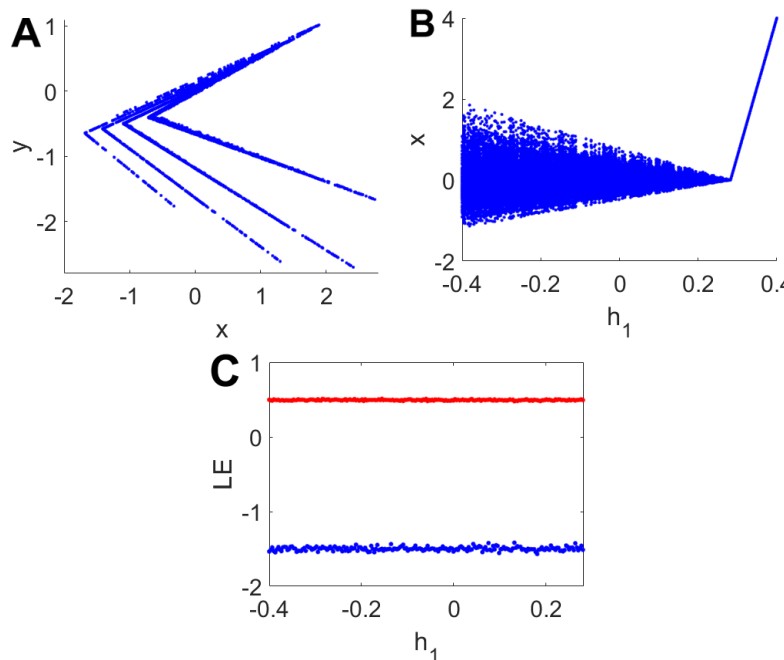

Figure 8: $A$) Phase portrait of a chaotic attractor due to a homoclinic interaction, $B$) bifurcation diagram, $C$) Lyapunov exponents (LEs). Parameter settings: $a_r = 1.5$, $b_r = -0.75$, $a_l = -1.77$, $b_l = -0.9$, $c = 0.6$, $d = 0.15$, $h_2 = -0.4$, $h_1 = -0.7$.

map. This means none of the bifurcation points in Fig. 9 can be examined in the 2d PL normal form map.

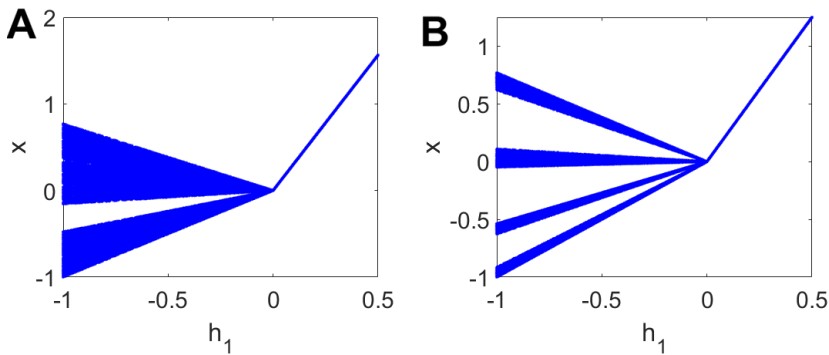

Figure 9: BCB diagrams for parameter values: $A$) $a_r = 0.68$, $b_r = -0.75$, $a_l = -1.77$, $b_l = -0.9$, $c = 0$, $d = 0.15$, $h_2 = -0.4$, $h_1 = -0.7$; and $B$) $a_r = 0.6$, $b_r = -0.75$, $a_l = -1.77$, $b_l = -0.9$, $c = 0$, $d = 0.15$, $h_2 = -0.4$, $h_1 = -0.7$.

### D.2.1 Homoclinic intersections for further iterations of fold points

In order to determine homoclinic intersections, it may be necessary to check further iterations of fold points for which we would need a recursive procedure. This will involve calculating the $n$-th power of a $2 \times 2$ matrix, and we therefore first prove the following Proposition:

**Proposition D.2.** *Let* $M = \begin{pmatrix} a & c \\ b & d \end{pmatrix}$ *have two distinct eigenvalues*

$$\lambda_{1,2} = \frac{a+d}{2} \mp \frac{\sqrt{(a-d)^2 + 4\,b\,c}}{2} = \frac{\Gamma}{2} \mp \frac{\sqrt{\Gamma^2 - 4\,\mathcal{D}}}{2}, \tag{23}$$

*where $\Gamma$ and $\mathcal{D}$ are the trace and determinant of $\boldsymbol{M}$. Then, for every $n \in \mathbb{N}$*

$$\boldsymbol{M}^n = \begin{pmatrix} A_{n+1} - d\,A_n & c\,A_n \\ b A_n & d\,A_n - \mathcal{D}\,A_{n-1} \end{pmatrix} \tag{24}$$

*where*

$$A_n = \frac{\lambda_1^n - \lambda_2^n}{\lambda_1 - \lambda_2}. \tag{25}$$

*Proof.* In our case $M$ is diagonalizable, so for $b \neq 0$

$$\boldsymbol{M} = \boldsymbol{V}\,\Lambda\,\boldsymbol{V}^{-1}$$

$$= \frac{1}{\lambda_1 - \lambda_2} \begin{pmatrix} \frac{\lambda_1 - d}{b} & \frac{\lambda_2 - d}{b} \\ 1 & 1 \end{pmatrix} \begin{pmatrix} \lambda_1 & 0 \\ 0 & \lambda_2 \end{pmatrix} \begin{pmatrix} b & d - \lambda_2 \\ -b & \lambda_1 - d \end{pmatrix}. \tag{26}$$

Therefore

$$\boldsymbol{M}^n = \boldsymbol{V}\,\Lambda^n\,\boldsymbol{V}^{-1}$$

$$= \frac{1}{\lambda_1 - \lambda_2} \begin{pmatrix} \frac{\lambda_1 - d}{b} & \frac{\lambda_2 - d}{b} \\ 1 & 1 \end{pmatrix} \begin{pmatrix} \lambda_1^n & 0 \\ 0 & \lambda_2^n \end{pmatrix} \begin{pmatrix} b & d - \lambda_2 \\ -b & \lambda_1 - d \end{pmatrix}$$

$$= \frac{1}{\lambda_1 - \lambda_2} \times$$

$$\begin{pmatrix} \lambda_2^n(d - \lambda_2) - \lambda_1^n(d - \lambda_1) & -\frac{(d - \lambda_1)(d - \lambda_2)}{b}(\lambda_1^n - \lambda_2^n) \\ b(\lambda_1^n - \lambda_2^n) & \lambda_1^n(d - \lambda_2) - \lambda_2^n(d - \lambda_1) \end{pmatrix}$$

$$= \begin{pmatrix} A_{n+1} - d\,A_n & -\frac{d^2 - \Gamma\,d + \mathcal{D}}{b}\,A_n \\ b A_n & d\,A_n - \mathcal{D}\,A_{n-1} \end{pmatrix}$$

$$= \begin{pmatrix} A_{n+1} - d\,A_n & c\,A_n \\ b A_n & d\,A_n - \mathcal{D}\,A_{n-1} \end{pmatrix}. \tag{27}$$

If $b = 0$, then the eigenvalues of $M$ are $\lambda_1 = a$ and $\lambda_2 = d$ ($a \neq d$). Thus,

$$\boldsymbol{M} = \boldsymbol{V}\,\Lambda\,\boldsymbol{V}^{-1} = \begin{pmatrix} 1 & \frac{-c}{a - d} \\ 0 & 1 \end{pmatrix} \begin{pmatrix} a & 0 \\ 0 & d \end{pmatrix} \begin{pmatrix} 1 & \frac{c}{a - d} \\ 0 & 1 \end{pmatrix}, \tag{28}$$

and so (for $n \in \mathbb{N}$)

$$\boldsymbol{M}^n = \boldsymbol{V}\,\Lambda^n\,\boldsymbol{V}^{-1} = \begin{pmatrix} 1 & \frac{-c}{a - d} \\ 0 & 1 \end{pmatrix} \begin{pmatrix} a^n & 0 \\ 0 & d^n \end{pmatrix} \begin{pmatrix} 1 & \frac{c}{a - d} \\ 0 & 1 \end{pmatrix}$$

$$= \begin{pmatrix} a^n & \frac{c(a^n - d^n)}{a - d} \\ 0 & d^n \end{pmatrix}, \tag{29}$$

which yields equation Eq. 24 for $b = 0$. $\qquad\square$

# E   Multi-stability

We next investigate multi-stability involving chaotic attractors. To do so, we need to obtain the regions of stability of various orbits and their overlap regions in the parameter space. In Sect. H.3, the existence of chaotic attractors was considered in a theoretical framework by finding necessary and sufficient conditions for the manifestation of homoclinic intersections. For $k$-periodic attractors ($k \geq 1$), the existence regions can be determined following a straightforward approach as briefly explained in Appx. G.

*Remark* E.1. The map Eq. 8 can exihit quasi-periodic orbits which are composed of an infinite number of points lying on an invariant closed curve, see Fig. 10. As illustrated in Fig. 10C, the largest Lyapunov exponent (red) is zero and the other one (blue) is negative which confirms the quasi-periodic orbit is stable. Multi-stability involving quasi-periodic attractors may therefore also be possible, but here we will focus on multi-stability of periodic and chaotic attractors.

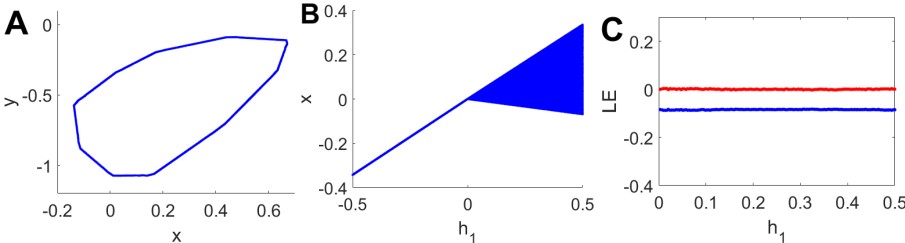

Figure 10: $A$) Phase portrait of a stable quasi-periodic orbit, $B$) bifurcation diagram, $C$) Lyapunov exponents (LEs). Parameter settings: $a_l = 0.31$, $b_l = -.6$, $a_r = -1.1$, $b_r = -1.54$, $,c = 0.9$, $d = 0.3$, $h_1 = 1$, $h_2 = 0$.

Finding the overlapping stability regions of different periodic orbits, we can observe various MABs which result in different multi-stabilities. In Fig. 11A the MAB diagram shows multi-stability of 2-cycles and 3-cycles after the bifurcation occurred. Fig. 11B illustrates multi-stability of $(i)$ 5-cycles and 2-cycles (before the bifurcation) and $(ii)$ 3-cycles and fixed points (after the bifurcation). Note that in both cases the term $(1 - d) h_1 + c h_2$ vanishes at the bifurcation point, so that none of the bifurcation points in Fig. 11 can be obtained through the 2d PL normal form map. Finding the

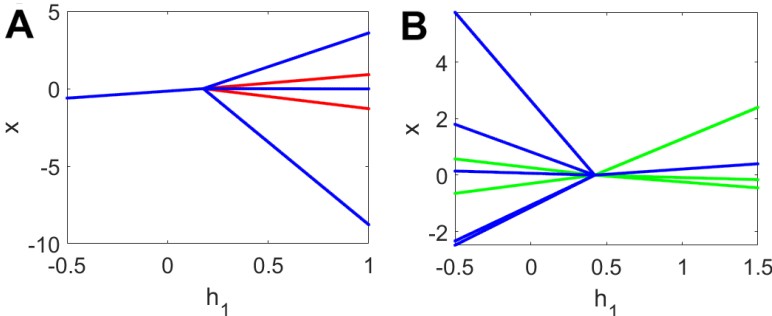

Figure 11: MAB diagrams for parameter settings: $A$) $c = 0.71$, $d = 0.2$, $b_l = -0.4$, $b_r = 0.5$, $a_l = 0.253$, $a_r = -2.83$, $h_2 = -0.2$; and $B$) $c = 0.95$, $d = 0.1$, $b_l = -1.21$, $b_r = -0.9$, $a_l = -2.98$, $a_r = -0.73$, $h_2 = -0.4$.

existence regions for homoclinic intersections (see Sect. H.3), we can investigate multi-stability involving chaotic attractors, as illustrated in the following examples. Suppose that all system parameters except $a_r$ have the same values as in Eq. 22. Fig. 12B represents multi-stability of a chaotic attractor and a stable 4-cycle for $a_r = 0.99$. Changing $h_1$ from negative to positive values, a multiple attractor bifurcation occurs at $h_1 \approx 0.282$. In Fig. 13B multi-stability of a 2-band chaotic attractor and a stable 3-cycle is apparent for $a_r = 0.3$. In this case, again, the system undergoes an MAB at $h_1 \approx 0.282$. Fig. 14 illustrates the coexistence of a chaotic and a 3-band chaotic attractor in two cases. As shown in Fig. 14A, there is an extended basin of attraction for the simple chaotic attractor whereas the 3-band attractor has a smaller basin. By contrast, the basins are more interwoven in Fig. 14B. Furthermore, in this case, one may consider the attractors in red as 2-band chaotic attractors based on the MAB diagram. Plotting the phase portrait, however, reveals they are 3-band chaotic attractors while projecting them onto either x or y axis yields 2-band attractors. Varying $h_1$ from negative to positive, the systems undergoes a MAB at which $(1 - d)h_1 + ch_2 = 0$.

Fig. 15B demonstrates the multi-stability of a chaotic attractor and a stable 3-cycle before a MAB. The bifurcation results in another type of multi-stability involving a stable fixed point and a 4-cycle.

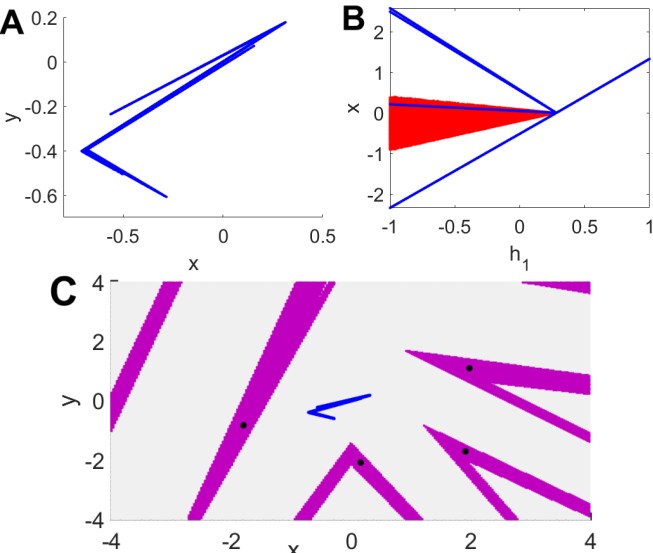

Figure 12: $A$) Phase portrait of a chaotic attractor, $B$) MAB diagram and multi-stability of a chaotic attractor (red) and a stable $4$-cycle (blue). $C$) Basins of attraction of the coexisting attractors. Parameter settings: $a_r = 0.99$, $b_r = -0.75$, $a_l = -1.77$, $b_l = -0.9$, $c = 0.6$, $d = 0.15$, $h_2 = -0.4$, $h_1 = -0.7$.

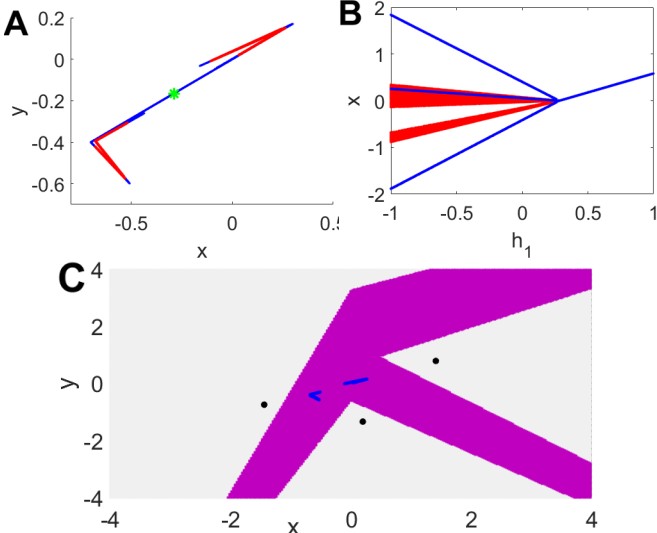

Figure 13: $A$) Unstable manifold (blue) of the saddle fixed point $\mathcal{O}_{\mathcal{L}}^* = (-0.28848, -0.16514)^{\mathsf{T}}$ (green star) and a 2-band chaotic attractor lying on it. $B$) MAB diagram and multi-stability of a 2-band chaotic attractor (red) and a stable $3$-cycle (blue). $C$) Basins of attraction of the coexisting attractors. Parameter settings: $a_r = 0.3$, $b_r = -0.75$, $a_l = -1.77$, $b_l = -0.9$, $c = 0.6$, $d = 0.15$, $h_2 = -0.4$, $h_1 = -0.7$.

## F  2-unit PLRNNs

Consider the 2-unit PLRNN defined by

$$
\boldsymbol{z}_t = F(\boldsymbol{z}_{t-1}) = (\boldsymbol{A} + \boldsymbol{W} \boldsymbol{D}_{\Omega(t-1)}) \boldsymbol{z}_{t-1} + \boldsymbol{h}
$$

$$
:= \boldsymbol{W}_{\Omega(t-1)} \, \boldsymbol{z}_{t-1} + \boldsymbol{h}, \tag{30}
$$

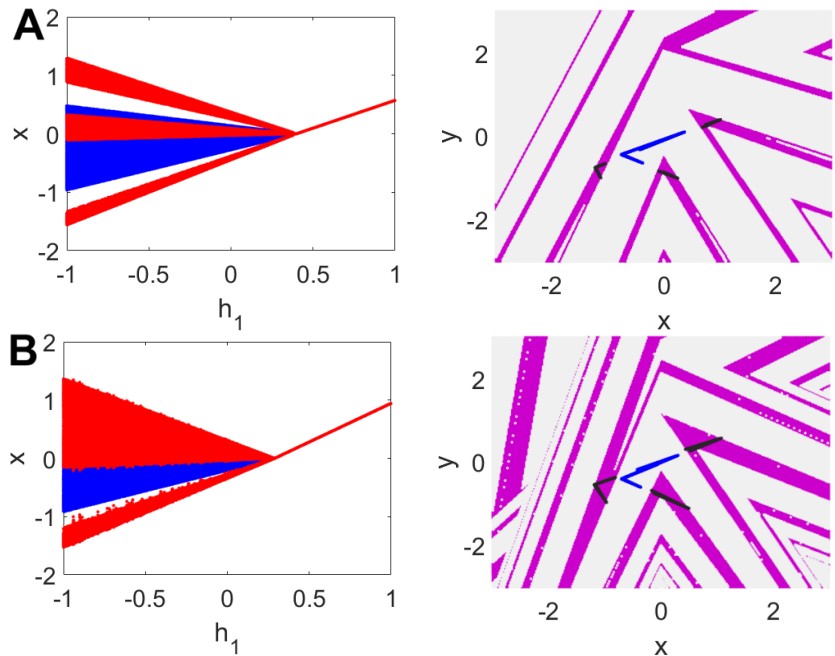

Figure 14: MAB diagram, multi-stability of a chaotic (blue) and a 3-band chaotic (red) attractor and their basins of attraction for parameter values: $A$) $a_r = 0.7$, $b_r = -0.75$, $a_l = -1.77$, $b_l = -0.9$, $c = 0.6$, $d = 0.4$, $h_2 = -0.4$, $h_1 = -0.7$; $B$) $a_r = 0.79$, $b_r = -0.75$, $a_l = -1.77$, $b_l = -0.9$, $c = 0.6$, $d = 0.17$, $h_2 = -0.4$, $h_1 = -0.7$.

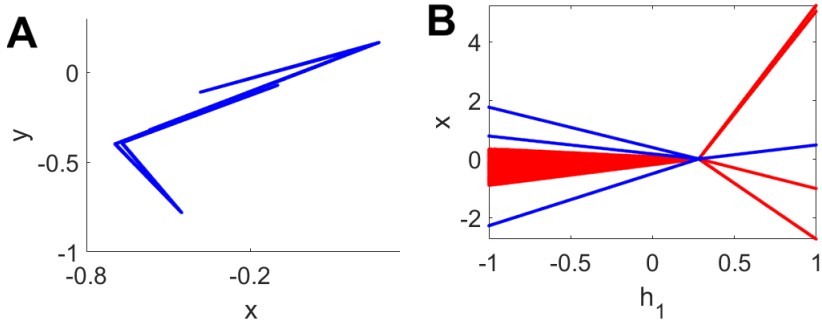

Figure 15: $A$) Phase portrait of a chaotic attractor, $B$) MAB diagram which shows the multi-stability of ($i$) a chaotic attractor (red) and a stable 3-cycle (blue) and ($ii$) a stable fixed point (blue) and a 4-cycle (red). Parameter settings: $a_r = 0.55$, $b_r = -1.5$, $a_l = -1.74$, $b_l = -0.9$, $c = 0.6$, $d = 0.15$, $h_2 = -0.4$, $h_1 = -0.7$.

where $\boldsymbol{z}_t = (z_{1t}, z_{2t})^\top \in \mathbb{R}^2$ indicates the neural state vector at time $t = 1 \cdots T$, the vector $\boldsymbol{h} = (h_1, h_2)^\top \in \mathbb{R}^2$ is the bias term, the matrices

$$\boldsymbol{A} = \begin{pmatrix} a_{11} & 0 \\ 0 & a_{22} \end{pmatrix}, \quad \boldsymbol{W} = \begin{pmatrix} w_{11} & w_{12} \\ w_{21} & w_{22} \end{pmatrix}, \tag{31}$$

consist of all (linear) auto-regression weights and connection weights respectively, $\boldsymbol{D}_{\Omega(t)} :=$ $\text{diag}(\boldsymbol{d}_{\Omega(t)})$ is a diagonal matrix and $\boldsymbol{d}_{\Omega(t)} := (d_1, d_2)$ an indicator vector with $d_m(z_{m,t}) := d_m = 1$ whenever $z_{m,t} > 0$, $m = 1, 2$, and zeros otherwise [15, 45]. Therefore, the phase space of system Eq. 30 is divided into 4 sub-regions by 4 hyper-surfaces as borders.Listing the 4 different configurations of $\boldsymbol{D}_{\Omega(t)}$ as $\boldsymbol{D}_{\Omega^k}$, $k = 1, 2, 3, 4$, we define 4 matrices

$$\boldsymbol{W}_{\Omega^k} := \boldsymbol{A} + \boldsymbol{W}\boldsymbol{D}_{\Omega^k}. \tag{32}$$

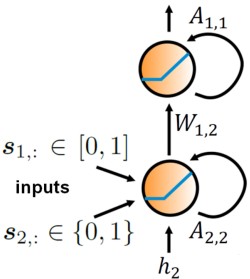

Figure 16: 2-unit PLRNN solution to addition problem [70].

In this case, in every indexed sub-region $S_{\Omega^k}$, $k = 1, 2, 3, 4$, the system dynamics are governed by a different map as follows

$$z_{t+1} = F(z_t) = W_{\Omega^k} z_t + h, \qquad z_t \in S_{\Omega^k}. \tag{33}$$

Using the binary number system, all the sub-regions $S_{\Omega^k}$'s can be defined as [53, 54]

$$S_{\Omega^1} = \hat{S}_0 = \hat{S}_{(0\,0)_2^*} = \hat{S}_{0\,0} = \left\{ z_t \in \mathbb{R}^2;\, z_{1t}, z_{2t} \leq 0 \right\},$$

$$S_{\Omega^2} = \hat{S}_1 = \hat{S}_{(0\,1)_2^*} = \hat{S}_{1\,0} = \left\{ z_t \in \mathbb{R}^2;\, z_{1t} > 0, z_{2t} \leq 0 \right\},$$

$$S_{\Omega^3} = \hat{S}_2 = \hat{S}_{(1\,0)_2^*} = \hat{S}_{0\,1} = \left\{ z_t \in \mathbb{R}^2;\, z_{2t} > 0, z_{1t} \leq 0 \right\},$$

$$S_{\Omega^4} = \hat{S}_3 = \hat{S}_{(1\,1)_2^*} = \hat{S}_{1\,1} = \left\{ z_t \in \mathbb{R}^2;\, z_{1t}, z_{2t} > 0 \right\}, \tag{34}$$

where each subindex $d$ of $\hat{S}$, $0 \leq d \leq 3$, is associated with a sequence $d_2\, d_1$ of binary digits. The notation $(d_1\, d_2)_2^*$ in building each corresponding sequence stands for the mirror image of the binary representation of $d$ with $M$ digits. By mirror image here we mean writing digits $d_1\, d_2$ from right to left, i.e. $d_2\, d_1$. Denoting switching boundaries $\Sigma_{ij} = \bar{S}_{\Omega^i} \cap \bar{S}_{\Omega^j}$ between every pair of successive sub-regions $S_{\Omega^i}$ and $S_{\Omega^j}$ with $i, j \in \{1, 2, 3, 4\}$, we can rewrite map Eq. 30 as

$$z_{t+1} = F(z_t) = \begin{cases} F_1(z_t) = W_{\Omega^1} z_t + h; & z_t \in \bar{S}_{\Omega^1} \\ F_2(z_t) = W_{\Omega^2} z_t + h; & z_t \in \bar{S}_{\Omega^2} \\ F_3(z_t) = W_{\Omega^3} z_t + h; & z_t \in \bar{S}_{\Omega^3} \\ F_4(z_t) = W_{\Omega^4} z_t + h; & z_t \in \bar{S}_{\Omega^4} \end{cases}. \tag{35}$$

If we consider the matrix $W$ in Eq. 31 as

$$W = \begin{pmatrix} w_{11} & 0 \\ w_{21} & 0 \end{pmatrix}, \tag{36}$$

then, applying definition Eq. 32, we have

$$W_{\Omega^1} = W_{\Omega^3} = \begin{pmatrix} a_{11} & 0 \\ 0 & a_{22} \end{pmatrix} = A,$$

$$W_{\Omega^2} = W_{\Omega^4} = \begin{pmatrix} a_{11} + w_{11} & 0 \\ w_{21} & a_{22} \end{pmatrix}. \tag{37}$$

In this case, there are only two different sub-regions divided by one border. That means considering $a_{11} = a_l$, $a_{11} + w_{11} = a_r$, $w_{21} = b_r$, $a_{22} = d$, the system Eq. 35 can be written as a 2-dimensional PL map with one switching boundary. On the other hand, studying generic 2d PL maps, we can investigate dynamics of 2-unit PLRNNs given by Eq. 35 locally near one border.

# G  Stability regions of $k$-cycles for the map Eq. 8

1065 The following is a concise description of how to determine the stability regions of $k$-cycles for
1066 $k = 1, 2, 3$. For sake of clarity, we focus on basic orbits. The regions of stability of other $k$-periodic
1067 orbits can be obtained similarly.

1068

1069 The fixed points of the map Eq. 8 are given by

$$\mathcal{O}_{\mathcal{L}/\mathcal{R}} =$$

$$\left( \frac{(1-d)\,h_1 + c\,h_2}{(1-d)(1-a_{l/r}) - b_{l/r}\,c}, \; \frac{b_{l/r}\,h_1 + (1-a_{l/r})\,h_2}{(1-d)(1-a_{l/r}) - b_{l/r}\,c} \right)^{\mathsf{T}}. \tag{38}$$

1070 $\mathcal{O}_{\mathcal{L}}$ and $\mathcal{O}_{\mathcal{R}}$ are admissible fixed points iff $x_{\mathcal{L}} < 0$ and $x_{\mathcal{R}} > 0$ respectively; otherwise they are
1071 virtual. Accordingly, the existence regions of the fixed points $\mathcal{O}_{\mathcal{L}}$ and $\mathcal{O}_{\mathcal{R}}$ are given by

$$E_{\mathcal{O}_{\mathcal{L}}} = \left\{ (h_1, h_2, a_l, b_l, c, d) \, \Big| \; \frac{(1-d)\,h_1 + c\,h_2}{(1-d)(1-a_l) - b_l\,c} < 0 \right\},$$

$$E_{\mathcal{O}_{\mathcal{R}}} = \left\{ (h_1, h_2, a_r, b_r, c, d) \, \Big| \; \frac{(1-d)\,h_1 + c\,h_2}{(1-d)(1-a_r) - b_r\,c} > 0 \right\}. \tag{39}$$

1072 Let $D_{\mathcal{L}/\mathcal{R}}$ and $\mathcal{P}_{\mathcal{L}/\mathcal{R}}(\pm 1)$ be the determinants and characteristic polynomials of the Jacobian matrices.
1073 Then, considering the conditions $\mathcal{P}_{\mathcal{L}/\mathcal{R}}(\pm 1) > 0$ and $D_{\mathcal{L}/\mathcal{R}} < 1$, the stability region of fixed points
1074 can be derived as

$$\mathcal{S}_{\mathcal{L}/\mathcal{R}} = \Big\{ (h_1, h_2, a_{l/r}, b_{l/r}, c, d) \in E_{\mathcal{O}_{\mathcal{L}/\mathcal{R}}} \, \big| \, a_{l/r}d - b_{l/r}c < 1,$$

$$1 \pm (a_{l/r} + d) + a_{l/r}\,d - b_{l/r}\,c > 0 \Big\}. \tag{40}$$

1075
1076 Likewise, the stability region of the 2-cycle $\mathcal{O}_{\mathcal{R}\mathcal{L}}$ is

$$\mathcal{S}_{\mathcal{R}\mathcal{L}} = \Big\{ (h_1, h_2, a_{l/r}, b_{l/r}, c, d) \in E_{\mathcal{O}_{\mathcal{R}\mathcal{L}}} \, \big| \; -1 <$$

$$(a_r\,d - b_r\,c)(a_l\,d - b_l\,c) < 1,$$

$$-(a_r\,d - b_r\,c)(a_l\,d - b_l\,c) - 1 <$$

$$c(b_l + b_r) + d^2 + a_l\,a_r <$$

$$(a_r\,d - b_r\,c)(a_l\,d - b_l\,c) + 1 \Big\}, \tag{41}$$

1077 in which

$$E_{\mathcal{O}_{\mathcal{R}\mathcal{L}}} = \Big\{ (h_1, h_2, a_l, b_l, c, d) \, \big|$$

$$\frac{\big((1-d)h_1 + c\,h_2\big)\big(a_l + d + a_l\,d - b_l\,c + 1\big)}{(a_r\,d - b_r\,c)(a_l\,d - b_l\,c) - c(b_l + b_r) - d^2 - a_l\,a_r + 1} > 0,$$

$$\frac{\big((1-d)h_1 + c\,h_2\big)\big(a_r + d + a_r\,d - b_r\,c + 1\big)}{(a_r\,d - b_r\,c)(a_l\,d - b_l\,c) - c(b_l + b_r) - d^2 - a_l\,a_r + 1} < 0 \Big\}, \tag{42}$$

1078 represents the existence region of $\mathcal{O}_{\mathcal{R}\mathcal{L}}$.
1079
1080 Analogously, the existence region of the 3-cycle $\mathcal{O}_{\mathcal{R}\mathcal{L}^2}$ is given by

$$\mathcal{S}_{\mathcal{R}\mathcal{L}^2} = \Big\{ (h_1, h_2, a_{l/r}, b_{l/r}, c, d) \in E_{\mathcal{O}_{\mathcal{R}\mathcal{L}^2}} \, \big|$$

$$-1 < (a_l\, d - b_l\, c)^2(a_r\, d - b_r\, c) < 1,$$
$$-(a_l\, d - b_l\, c)^2(a_r\, d - b_r\, c) - 1 < a_l^2\, a_r + d^3$$
$$+ c\big(a_l\, b_l + a_l\, b_r + a_r\, b_l + d(2\, b_l + b_r)\big) <$$
$$(a_l\, d - b_l\, c)^2(a_r\, d - b_r\, c) + 1 \Big\}, \tag{43}$$

where

$$E_{\mathcal{O}_{\mathcal{RL}^2}} = \Big\{ (h_1, h_2, a_l, b_l, c, d) \,\big|\, \frac{\big((1-d)h_1 + c\, h_2\big)G_1}{G} > 0,$$
$$\frac{\big((1-d)h_1 + c\, h_2\big)K_1}{G} < 0, \ \frac{\big((1-d)h_1 + c\, h_2\big)H_1}{G} < 0 \Big\}, \tag{44}$$

and

$$G_1 = a_l^2\, d^2 + a_l^2\, d + a_l^2 - 2\, a_l\, b_l\, c\, d - a_l\, b_l\, c + a_l\, d^2 + a_l\, d$$
$$+ a_l + b_l^2\, c^2 - b_l\, c\, d + b_l\, c + d^2 + d + 1,$$
$$G = -a_l^2\, a_r - d^3 - c\big(a_l\, b_l + a_l\, b_r + a_r\, b_l + d(2\, b_l + b_r)\big)$$
$$+ (a_r\, d - b_r\, c)(a_l\, d - b_l\, c)^2 + 1,$$
$$K_1 = a_r + d + a_l a_r + b_l c + a_r d + a_r d^2 + d^2 + a_l a_r\, d - a_l b_r c$$
$$- b_r\, c\, d + a_l\, a_r\, d^2 + b_l\, b_r\, c^2 - a_l\, b_r\, c\, d - a_r\, b_l\, c\, d + 1,$$
$$H_1 = a_l + d + a_l a_r + a_l d + b_r c + a_l d^2 + d^2 + a_l a_r d - a_r b_l c$$
$$- b_l\, c\, d + a_l\, a_r\, d^2 + b_l\, b_r\, c^2 - a_l\, b_r\, c\, d - a_r\, b_l\, c\, d + 1. \tag{45}$$

# H  Algorithms

## H.1  Methodological details

All experiments were run on a single CPU, specifically an Intel Xeon Gold 6132 with 512GB RAM and an Intel Xeon Gold 6248 with 832GB RAM.

| Parameter | Lorenz63 Fig. 2A | Oscillator Fig. 2B | Lorenz63 Fig. 2C | Duffing Fig. 4A | Decision making Fig. 4B | Lorenz63 Fig. 4C | Empirical Fig. 4D |
|---|---|---|---|---|---|---|---|
| Model | ALRNN | ALRNN | ALRNN | shallowPLRNN | ALRNN | shallowPLRNN | ALRNN |
| Latent dim | 10/20/30/50 | 40 | 30 | 2 | 15 | 3 | 25 |
| Hidden dim | - | - | - | 10 | - | 20 | - |
| #ReLUs | Latent dim - 3 | 15 | 8 | - | 6 | - | 6 |
| Sequence length | 200 | 25 | 100 | 100 | 100 | 100 | 200 |
| Gaussian noise | 0.0 | 0.0 | 0.0 | 0.0 | 0.01 | 0.05 | 0.02 |
| $\lambda_{\text{invert}}$ | 0.0/0.1·exp(Latent dim) | 0.0/1e15 | 0.0/1e10 | 0.0 | 0.2 | 0.0 | 0.3 |
| Batch Size | 16 | 16 | 16 | 32 | 16 | 16 | 16 |
| Epochs | 10000 | 1000 | 1000 | 10000 | 20000 | 1000 | 2000 |
| Start LR | 0.001 | 0.001 | 0.005 | 0.001 | 0.005 | 0.005 | 0.004 |
| TF interval | 16 | 10 | 15 | 15 | 15 | 15 | 20 |

Table 3: Parameter configurations for the different experiments. TF = teacher forcing. LR = learning rate.

For Fig. 2A, every 500 epochs 10 latent trajectories of length 1000 were produced to determine the pool of linear subregions $\mathcal{S}$ traversed by the model. To exclude transients, the first 99 time steps were discarded from each trajectory. The regularized linear subregions $\mathcal{S}_{\text{reg}}$ were chosen at random as a fraction (e.g. 1%, 10%, depending on the hyperparameter) from this pool $\mathcal{S}$ for each epoch. To evaluate the invertibility ($\mathcal{S}_+/\mathcal{S}$), 20 trajectories of length 10000 were generated and a pool of subregions computed from. The first 99 time steps were again discarded to remove possible transients.

## H.2 Additional algorithms

In the main text, the algorithm was formulated for the PLRNN. The same procedure as described in Algo. 2 can analogously be applied to a 1-hidden layer PLRNN, called the shallow PLRNN (shPLRNN) [36]:

$$z_{t+1} = Az_t + W_1 D_t(W_2 z_t + h_2) + h_1 \tag{46}$$

where $A \in \mathbb{R}^{M \times M}$, $W_1 \in \mathbb{R}^{M \times H}$, $W_2 \in \mathbb{R}^{H \times M}$, $h_2 \in \mathbb{R}^H$, $h_1 \in \mathbb{R}^M$ and $H > M$. The inversion of this map yields

$$z_{t-1} = (A + W_1 D_{t-1} W_2)^{-1}(z_t - W_1 D_{t-1} h_2 - h_1) \tag{47}$$

**Fallback algorithm** For systems with complex dynamics, such as in Fig. 3B where trajectories jump between disjoint subregions, the primary algorithm may struggle to converge. In such cases, we use a more robust fallback method (Alg. 3) that perturbs seed points along the analytically defined local manifold and iterates $F_\theta$ to generate dense support vectors. As manifolds can re-enter the same subregion in multiple folds, we apply HDBSCAN [51, 20] to cluster support vectors into distinct segments. Although computationally more demanding, this fallback reliably captures manifolds with discontinuous or folding structures when sequential tracing fails.

---

**Algorithm 2** Backtracking Time Series in a ReLU based RNNs

---

1: $z_T \leftarrow$ an initial State
2: $\theta \leftarrow$ Parameters
3: Initialize list: $S = [z_T]$
4: **for** $t = T : 1$ **do**
5:      $z_t = S[T - t]$
6:      $D_{t-1} \leftarrow \text{diag}(z_t > 0)$             ▷ Initialize D as a diagonal matrix
7:      $z_{t-1}^* \leftarrow F^{-1}(\theta, D_{t-1}, z_t)$             ▷ Perform a backward step
8:      $\tilde{z}_t \leftarrow F(\theta, z_{t-1}^*)$             ▷Perform a forward step
9:      **if** $\tilde{z}_t = z_t$ **then**
10:          $S \leftarrow S \cup \{z_{t-1}^*\}$             ▷If forward step is correct
11:      **else**
12:          $D_{t-1} \leftarrow \text{diag}(z_{t-1}^* > 0)$             ▷Update D with new candidate
13:          $z_{t-1}^* \leftarrow F^{-1}(\theta, D_{t-1}, z_t)$             ▷ Retry backward step
14:          $\tilde{z}_t \leftarrow F(\theta, z_{t-1}^*)$             ▷ Retry forward step
15:          **if** $\tilde{z}_t = z_t$ **then**
16:              $S \leftarrow S \cup \{z_{t-1}^*\}$             ▷ If forward step is correct
17:          **else**
18:              $\tilde{z}_t \leftarrow$ TryPreviousRegions$(\theta, D\_pool, z_t, z_{t-1}^*, \tilde{z}_t)$      ▷ Try previous regions
19:              **if** $\tilde{z}_t \neq z_t$ **then**
20:                  $\tilde{z}_t \leftarrow$ TryBitflips$(\theta, z_t, z_{t-1}^*, \tilde{z}_t)$      ▷ Hierarchically check neighbours
21:                  **if** $\tilde{z}_t = z_t$ **then**
22:                      $S \leftarrow S \cup \{z_{t-1}^*\}$
23:                  **else**
24:                      **return**
25:                  **end if**
26:              **end if**
27:          **end if**
28:      **end if**
29: **end for**
30: **return** trajectory

---

**Algorithm 3** Finding stable/unstable manifolds: fallback algorithm

1: $(P, E) \leftarrow$ SCYFI         ▷P: Fixed Point, E: Eigenvectors
2: $S^s \leftarrow \emptyset$         ▷Stable Manifolds
3: $S^u \leftarrow \emptyset$         ▷Unstable Manifolds
4: **for** i=1:$N_1$ **do**
5:     $z_0 = P$         ▷For $N_1$ different initialisations
6:     **for** $v^u \in E^u$ **do**
7:         $z_0 \mathrel{+}= v^u \cdot rand()$         ▷Perturbe into subspace
8:     **end for**
9:     $T^u \leftarrow GetForwardTS(z_0)$
10:     $S^u \leftarrow S^u \cup \{T^u\}$
11: **end for**
12: **for** i=1:$N_2$ **do**
13:     $z_0 = P$         ▷ For $N_2$ different initialisations
14:     **for** $v^s \in E^s$ **do**
15:         $z_0 \mathrel{+}= v^s \cdot rand()$         ▷ Perturb into subspace
16:     **end for**
17:     $T^s \leftarrow GetBackwardTS(z_0)$
18:     $S^s \leftarrow S^s \cup \{T^s\}$
19: **end for**
20: $\widetilde{S}^s \leftarrow \emptyset$         ▷ Piecewise linear manifold fits
21: $\widetilde{S}^u \leftarrow \emptyset$
22: **for each** $D \in D_\Omega$ **do**
23:     $S^s_\Omega \leftarrow S^s \cap D_\Omega$         ▷ Go through all subregions
24:     $S^u_\Omega \leftarrow S^u \cap D_\Omega$
25:     $(C^s_\Omega, C^u_\Omega) \leftarrow$ FIT$((S^s_\Omega, S^u_\Omega))$         ▷ Cluster points and fit
26:     $\widetilde{S}^s \leftarrow \widetilde{S}^s \cup \{C^s_\Omega\}$
27:     $\widetilde{S}^u \leftarrow \widetilde{S}^u \cup \{C^u_\Omega\}$
28: **end for**
29: **return** $(\widetilde{S}^s, \widetilde{S}^u)$         ▷ Piecewise linear manifolds

---

1: **function** BACKWARDFORWARD$(\Theta, D, z)$
2:     $z^* \leftarrow F{-}1(\Theta, D, z)$
3:     $\tilde{z} \leftarrow F(\Theta, z^*)$
4:     **return** $z^*, \tilde{z}$
5: **end function**
6: **function** TRYPREVIOUSREGIONS$(\Theta, D\_pool, z, z^*, \tilde{z})$
7:     **for** $D \in D\_pool$ **do**
8:         $z^*, \tilde{z} \leftarrow$ BackwardForward$(\Theta, D, z)$
9:         **if** $\tilde{z} = z$ **then**
10:             **return** $z^*$
11:         **end if**
12:     **end for**
13: **end function**
14: **function** TRYBITFLIPS$(\Theta, z, z^*, \tilde{z})$
15:     **for** $k = 1 : num\_relus$ **do**
16:         $D\_versions \leftarrow$ generate_bitflip_k()
17:         **for** $D \in D\_versions$ **do**
18:             $z^*, \tilde{z} \leftarrow$ BackwardForward$(\Theta, D, z)$
19:             **if** $\tilde{z} = z$ **then**
20:                 **return** $z^*$
21:             **end if**
22:         **end for**
23:     **end for**
24: **end function**

### H.3 Existence of homoclinic intersections

A horseshoe structure is generated when there is a homoclinic intersection between stable and unstable manifolds of a saddle fixed point and therefore an infinite number of intersections. The occurrence of a homoclinic intersection implies the existence of a chaotic orbit [82]. Here we will obtain a necessary and sufficient condition for the occurrence of homoclinic intersections in order to find a general condition for the existence of chaos. For this we consider the system Eq. 8 where $T$ is invertible and the matrices $\boldsymbol{A}_{\mathcal{L}}$ and $\boldsymbol{A}_{\mathcal{R}}$ are non-singular and have no eigenvalue equal to 1. Note that as the parameter space of general 2d maps is eight-dimensional, the computation of manifolds could be more challenging than for 2d normal form maps. We approach this issue by establishing the conditions of homoclinic intersections using the equations of the stable and unstable eigenlines.

#### H.3.1 Analytical condition for homoclinic intersection of the first and second fold points

Let $\mathcal{O}_{\mathcal{L}}^{*} = (x_{\mathcal{L}}^{*}, y_{\mathcal{L}}^{*})^{\mathsf{T}}$ be an admissible saddle fixed point in the left sub-region $\mathcal{L}$. Since $\mathcal{O}_{\mathcal{L}}^{*}$ is a saddle, $\boldsymbol{A}_{\mathcal{L}}$ has one stable and one unstable eigenvalue $\lambda_s$ and $\lambda_u$, respectively. Let us denote the line generated by the associated stable eigenvector by $\boldsymbol{\ell}_s^{*}$ and the line produced by the corresponding unstable eigenvector by $\boldsymbol{\ell}_u^{*}$. Since the unstable eigenvector is $\boldsymbol{v}_u = (v_1, v_2)^{\mathsf{T}} = (\frac{\lambda_u - d}{b_l}, 1)^{\mathsf{T}}$, $\boldsymbol{\ell}_u^{*}$ hits the border $x = 0$ at $P_0 = (0, y_0)^{\mathsf{T}} \in \Sigma$ where

$$
\begin{aligned}
y_0 &= y_{\mathcal{L}}^{*} - x_{\mathcal{L}}^{*} \frac{v_2}{v_1} \\
&= \frac{b_l\, h_1 + (1 - a_l)\, h_2}{1 - \Gamma_{\mathcal{L}} + D_{\mathcal{L}}} - \frac{(1 - d)\, h_1 + c\, h_2}{1 - \Gamma_{\mathcal{L}} + D_{\mathcal{L}}} \left( \frac{b_l}{\lambda_u - d} \right) \\
&= \frac{\big(b_l\, h_1 + (1 - a_l)h_2\big)(\lambda_u - d) - b_l\big(c\, h_2 + (1 - d)h_1\big)}{(\lambda_u - d)(1 - \Gamma_{\mathcal{L}} + D_{\mathcal{L}})}.
\end{aligned}
\tag{48}
$$

The image of $P_0$ is the first fold point of the unstable manifold of $\mathcal{O}_{\mathcal{L}}^{*}$, and so all its images will also be fold points. Its coordinate is $P_1 = (c\, y_0 + h_1, d\, y_0 + h_2)^{\mathsf{T}}$. The image of the first fold point is the second fold point $P_2 = (x_2, y_2)^{\mathsf{T}}$ with coordinates

$$
\begin{cases}
x_2 = c(a_l + d)y_0 + (a_l + 1)h_1 + ch_2 \\
y_2 = (b_l c + d^2)y_0 + b_l h_1 + (d + 1)h_2
\end{cases}
, \quad \text{if } c\, y_0 + h_1 < 0
$$

$$
\begin{cases}
x_2 = c(a_r + d)y_0 + (a_r + 1)h_1 + ch_2 \\
y_2 = (b_r c + d^2)y_0 + b_r h_1 + (d + 1)h_2
\end{cases}
, \quad \text{if } c\, y_0 + h_1 > 0
\tag{49}
$$

Now we check whether or not the points $P_1$ and $P_2$ are on opposite sides of the stable eigenline $\boldsymbol{\ell}_s^{*}$. When $P_1$ and $P_2$ are on opposite sides of $\boldsymbol{\ell}_s^{*}$, then the unstable manifold must have intersected the stable manifold. Thus, we have a homoclinic intersection which implies the occurrence of chaotic dynamics. Since the stable eigenvector is $\boldsymbol{v}_s = (\frac{\lambda_s - d}{b_l}, 1)^{\mathsf{T}}$ and the eigenline $\boldsymbol{\ell}_s^{*}$ passes through $\mathcal{O}_{\mathcal{L}}^{*} = (x_{\mathcal{L}}^{*}, y_{\mathcal{L}}^{*})^{\mathsf{T}}$, $\boldsymbol{\ell}_s^{*}$ can be computed as

$$
\begin{aligned}
\boldsymbol{\ell}_s^{*}: \quad & \frac{\lambda_s - d}{b_l} y - x + \frac{(1 - d)\, h_1 + c\, h_2}{1 - \Gamma_{\mathcal{L}} + D_{\mathcal{L}}} \\
& - \left( \frac{\lambda_s - d}{b_l} \right) \frac{b_l\, h_1 + (1 - a_l)\, h_2}{1 - \Gamma_{\mathcal{L}} + D_{\mathcal{L}}} =: \mathsf{L}(x, y) = 0.
\end{aligned}
\tag{50}
$$

Now there are two possibilities:

**Case I**: $\mathsf{L}(x_1, y_1) \cdot \mathsf{L}(x_2, y_2) \leq 0$

If $\mathsf{L}(x_1, y_1) \cdot \mathsf{L}(x_2, y_2) < 0$, then $P_1$ and $P_2$ are on opposite sides of the stable manifold, while $\mathsf{L}(x_1, y_1) \cdot \mathsf{L}(x_2, y_2) = 0$ implies that at least one of the points $P_1$ and $P_2$ lies exactly on the stable

manifold. In both cases there exists a homoclinic intersection, i.e., whenever we have

$$
\left( \frac{\lambda_s - d}{b_l} (dy_0 + h_2) - (cy_0 + h_1) + \mathcal{M} \right) \left( \frac{\lambda_s - d}{b_l} \big( (b_{l/r}c + d^2)y_0 \right.
$$

$$
+ b_{l/r}h_1 + (d+1)h_2 \big) - \big( c(a_{l/r} + d)y_0 + (a_{l/r} + 1)h_1 + ch_2 \big)
$$

$$
\left. + \mathcal{M} \right) \le 0, \tag{51}
$$

where $\mathcal{M} = \frac{(1-d)\,h_1 + c\,h_2}{1 - \Gamma_{\mathcal{L}} + D_{\mathcal{L}}} - \left( \frac{\lambda_s - d}{b_l} \right) \frac{b_l\,h_1 + (1 - a_l)\,h_2}{1 - \Gamma_{\mathcal{L}} + D_{\mathcal{L}}}$.

The homoclinic intersection point $P_{hom} = (x_{hom}, y_{hom})^{\mathsf{T}}$ is the point of intersection between the line joining the two fold points and the stable eigenline $\ell_s^*$, and hence given by

$$
\begin{cases} x_{hom} = (x_2 - x_1)\beta + x_1 \\ y_{hom} = (y_2 - y_1)\beta + y_1 \end{cases}, \tag{52}
$$

where

$$
\beta = \frac{x_1 - \frac{\lambda_s - d}{b_l} - \mathcal{M}}{\frac{\lambda_s - d}{b_l}(y_2 - y_1) - (x_2 - x_1)}. \tag{53}
$$

Finally, we must ensure that the intersection happens before the stable eigenline hits the border. For this, we need to have $x_{home}\, x_{\mathcal{L}}^* > 0$, which implies the intersection point $P_{hom}$ and the fixed point $\mathcal{O}_{\mathcal{L}}^*$ are on the same side of $\Sigma$.

**Case II**: $\mathsf{L}(x_1, y_1) \cdot \mathsf{L}(x_2, y_2) > 0$

If $\mathsf{L}(x_1, y_1) \cdot \mathsf{L}(x_2, y_2) > 0$, we need to check whether the unstable manifold intersects with the part of the stable manifold which ensues after folding along the $y$-axis. For this we have to calculate more points of the global stable manifold. Since $T$ is assumed to be invertible, the global stable manifold is formed by the union of all preimages (inverses) of any rank of the local stable set (a segment of the local stable eigenline). Assume, under the action of $T^{-1}$, the line $\ell_s^*$ maps to the $y$-axis and intersects it at $\tilde{P}_0 = (0, \tilde{y}_0)^{\mathsf{T}} \in \Sigma$. Then $\tilde{P}_0$ is the first fold point of the stable manifold of $\mathcal{O}_{\mathcal{L}}^*$, and $\tilde{y}_0$ is given by

$$
\tilde{y}_0 = \frac{\big( b_l\, h_1 + (1 - a_l)h_2 \big)(\lambda_s - d) - b_l\big( c\, h_2 + (1 - d)h_1 \big)}{(\lambda_s - d)(1 - \Gamma_{\mathcal{L}} + D_{\mathcal{L}})}. \tag{54}
$$

The preimage of $\tilde{P}_0$ is the second fold point $\tilde{P}_{-1} = (\tilde{x}_{-1}, \tilde{y}_{-1})^{\mathsf{T}}$ with coordinates

$$
\begin{cases} \tilde{x}_{-1} = \frac{1}{D_{\mathcal{L}}}\big( -c\tilde{y}_0 + ch_2 - dh_1 \big) \\ \tilde{y}_{-1} = \frac{1}{D_{\mathcal{L}}}\big( a_l\tilde{y}_0 + b_l h_1 - a_l h_2 \big) \end{cases}, \quad \text{if } \frac{\varphi^{\mathsf{T}}(\tilde{P}_0 - \boldsymbol{h})}{D_{\mathcal{L}}} \le 0
$$

$$
\begin{cases} \tilde{x}_{-1} = \frac{1}{D_{\mathcal{R}}}\big( -c\tilde{y}_0 + ch_2 - dh_1 \big) \\ \tilde{y}_{-1} = \frac{1}{D_{\mathcal{R}}}\big( a_r\tilde{y}_0 + b_r h_1 - a_r h_2 \big) \end{cases}, \quad \text{if } \frac{\varphi^{\mathsf{T}}(\tilde{P}_0 - \boldsymbol{h})}{D_{\mathcal{R}}} \ge 0 \tag{55}
$$

where

$$
\frac{\varphi^{\mathsf{T}}(\tilde{P}_0 - \boldsymbol{h})}{D_{\mathcal{L}/\mathcal{R}}} = \frac{-d\, h_1}{D_{\mathcal{L}/\mathcal{R}}} - \frac{c}{D_{\mathcal{L}/\mathcal{R}}}(\tilde{y}_0 - h_2). \tag{56}
$$

Now the line joining the two fold points $\tilde{P}_0$ and $\tilde{P}_{-1}$ is given by $\tilde{\mathsf{L}}(x, y) = 0$ where

$$
\tilde{\mathsf{L}}(x, y) := y - \tilde{y}_0 - \frac{\tilde{y}_{-1} - \tilde{y}_0}{\tilde{x}_{-1}}\, x. \tag{57}
$$

If $\tilde{\mathsf{L}}(x_1, y_1) \cdot \tilde{\mathsf{L}}(x_2, y_2) < 0$, then $P_1$ and $P_2$ are on opposite sides of the stable manifold, and the unstable manifold intersects the stable manifold at $\tilde{P}_{hom} = (\tilde{x}_{hom}, \tilde{y}_{hom})^{\mathsf{T}}$ with

$$
\begin{cases} \tilde{x}_{hom} = (x_2 - x_1)\tilde{\beta} + x_1 \\ \tilde{y}_{hom} = (y_2 - y_1)\tilde{\beta} + y_1 \end{cases}, \tag{58}
$$

where

$$\tilde{\beta} = \frac{(\tilde{y}_0 - y_1)\tilde{x}_{-1} + (\tilde{y}_{-1} - \tilde{y}_0)x_1}{(y_2 - y_1)\tilde{x}_{-1} - (\tilde{y}_{-1} - \tilde{y}_0)(x_2 - x_1)}. \tag{59}$$

We need to have $\tilde{x}_{home}\,\tilde{x}_{-1} > 0$, which means the intersection point $\tilde{P}_{hom}$ and the point $\tilde{P}_{-1}$ are on the same side of $\Sigma$.

*Remark* H.1. An analogous procedure can be performed to analytically obtain homoclinic intersections for the fixed point $\mathcal{O}_\mathcal{R}^* \in \mathcal{R}$.

## A recursive algorithm for determining homoclinic intersections

To investigate the existence of homoclinic intersections for the map Eq. 8, we use an algorithm similar to the one proposed in [67]. As illustrated in Fig. 17, it is based on a recursive procedure as follows:

1. Let $\mathcal{O}_\mathcal{L}^* = (x_\mathcal{L}^*, y_\mathcal{L}^*)^\mathsf{T} \in \mathcal{L}$ be an admissible saddle fixed point, and $\lambda_s$, $\lambda_u$ stable and unstable eigenvalues of $\boldsymbol{A}_\mathcal{L}$. Assume that $\boldsymbol{\ell}_s^*$ and $\boldsymbol{\ell}_u^*$ are the stable and unstable eigenlines generated by the associated stable and unstable eigenvectors, respectively. Analogous to Sect. H.3.1, suppose that $\boldsymbol{\ell}_u^*$ hits the border at $P_0 = (0, y_0)^\mathsf{T} \in \Sigma$ where $y_0$ is given by Eq. 48.

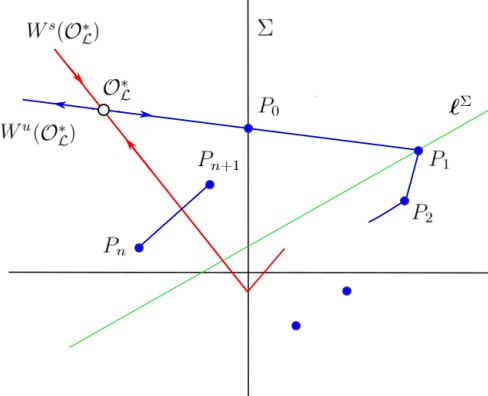

Figure 17: Schematic diagram to illustrate the procedure for finding homoclinic intersections.

2. Consider the image of $P_0$ and assume this point is on the right side of the border, i.e. $P_1 \in \mathcal{R}$. Since this point has the coordinate $P_1 = (c\,y_0 + h_1, d\,y_0 + h_2)^\mathsf{T}$, it follows that

$$c\,y_0 + h_1$$
$$= c\,\frac{\big(b_l\,h_1 + (1 - a_l)h_2\big)(\lambda_u - d) - b_l\big(c\,h_2 + (1 - d)h_1\big)}{(\lambda_u - d)(1 - \Gamma_\mathcal{L} + D_\mathcal{L})}$$
$$+ \,h_1 > 0. \tag{60}$$

Moreover $P_1 \in \ell^\Sigma$ is the first fold point of the unstable manifold of $\mathcal{O}_\mathcal{L}^*$, and so all its images will also be fold points.

3. Suppose that the orbit starting from $P_0$ will return to the border again. Let $n$ be the the border return time defined as the minimum number of iterations needed for $P_0$ to cross the border and return to the left side again, i.e., such that all iterations $P_1, P_2, \cdots, P_{n-1}$ lie on the right hand side while $P_n \in \mathcal{L}$. Using a recursive method similar to the one proposed in [67], we can compute the $n$-th iteration, $P_n$, directly from $P_0$. That is, there is no need to compute any other previous iterations $P_1, P_2, \cdots, P_{n-1}$. For this purpose, first we calculate $P_n$ as

$$P_n = T^n(P_0) = \boldsymbol{A}_\mathcal{R}^n\,P_0 + \big(\boldsymbol{A}_\mathcal{R} - \boldsymbol{I}\big)^{-1}\big(\boldsymbol{A}_\mathcal{R}^n - \boldsymbol{I}\big)\boldsymbol{h}. \tag{61}$$

Suppose that $\boldsymbol{A}_{\mathcal{R}}$ has two distinct eigenvalues, then, according to Proposition D.2, the matrix $\boldsymbol{A}_{\mathcal{R}}^n$ has the form Eq. 24. Hence, for $P_0 = (0, y_0)^{\mathsf{T}}$ we have

$$
P_n = \begin{pmatrix} A_{n+1} - d\,A_n & c\,A_n \\ b_r\,A_n & d\,A_n - \mathcal{D}_{\mathcal{R}}\,A_{n-1} \end{pmatrix} \begin{pmatrix} 0 \\ y_0 \end{pmatrix} + \frac{-1}{\mathcal{P}_{\mathcal{R}}(1)} \times
$$

$$
\begin{pmatrix} 1 - d & c \\ b_r & 1 - a_r \end{pmatrix} \begin{pmatrix} A_{n+1} - dA_n - 1 & cA_n \\ b_r A_n & d A_n - \mathcal{D}_{\mathcal{R}} A_{n-1} - 1 \end{pmatrix} \begin{pmatrix} h_1 \\ h_2 \end{pmatrix} =
$$

$$
\begin{pmatrix} cA_n\, y_0 - \dfrac{\left( (1-d)\left[ A_{n+1} - dA_n - 1 \right] + cb_r A_n \right) h_1 + c\left[ A_n - \mathcal{D}_{\mathcal{R}} A_{n-1} - 1 \right] h_2}{\mathcal{P}_{\mathcal{R}}(1)} \\[2ex] \left( dA_n - \mathcal{D}_{\mathcal{R}} A_{n-1} \right) y_0 - \dfrac{b_r \left[ A_n (1 - \Gamma_{\mathcal{R}}) + A_{n+1} - 1 \right] h_1 + b^* h_2}{\mathcal{P}_{\mathcal{R}}(1)} \end{pmatrix} \tag{62}
$$

where $A_n$ is given by Eq. 25; and $\Gamma_{\mathcal{R}}, \mathcal{D}_{\mathcal{R}}, \mathcal{P}_{\mathcal{R}}$ are the trace, determinant and characteristic polynomial of $\boldsymbol{A}_{\mathcal{R}}$ respectively; and

$$
b^* = (a_r - 1)\left( 1 + \mathcal{D}_{\mathcal{R}} A_{n-1} \right) + \left( d - \mathcal{D}_{\mathcal{R}} \right) A_n. \tag{63}
$$

Now, by Eq. 62 we can compute all iterations and thus find the border return time $n$ for which the unstable manifold passes the border again.

4. Finally, we calculate $P_{n+1}$ and check whether or not the points $P_{n+1}$ and $P_n$ are on opposite sides of the stable eigenline $\ell_s^*$, i.e. whether we obtain a homoclinic intersection. For this, let $P_n = (x_n, y_n)^{\mathsf{T}}$, $P_{n+1} = (x_{n+1}, y_{n+1})^{\mathsf{T}}$, and consider $\mathsf{L}(x, y)$ given by Eq. 50. If $\mathsf{L}(x_n, y_n) \cdot \mathsf{L}(x_{n+1}, y_{n+1}) < 0$, then $P_n$ and $P_{n+1}$ are on opposite sides of the stable manifold, while $\mathsf{L}(x_n, y_n) \cdot \mathsf{L}(x_{n+1}, y_{n+1}) = 0$ implies at least one of the points $P_n$ or $P_{n+1}$ lies exactly on the stable manifold. In both cases there exists a homoclinic intersection.

---

**Algorithm 4** Investigating Homoclinic Intersections in 2D

---

1: **procedure** INVESTIGATEHOMOCLINICINTERSECTIONS
2:    $\mathcal{O}_{\mathcal{L}}^* \leftarrow (x_{\mathcal{L}}^*, y_{\mathcal{L}}^*)^{\mathsf{T}}$               $\triangleright$ Admissible saddle fixed point
3:    $(\lambda_s, \lambda_u) \leftarrow$ (Stable Eigenvalue, Unstable Eigenvalue)
4:    $(\ell_s^*, \ell_u^*) \leftarrow$ (Stable Eigenline, Unstable Eigenline)
5:    $P_0 \leftarrow (0, y_0)^{\mathsf{T}} \in \Sigma$         $\triangleright$ Unstable eigenline hits the border at $P_0$
6:    $P_1 \leftarrow (cy_0 + h_1, dy_0 + h_2)^{\mathsf{T}}$       $\triangleright$ Image of $P_0$ on the right side of the border
7:    $P_1 \in \ell^{\Sigma}$               $\triangleright$ First fold point of the unstable manifold
8:    $n \leftarrow$ Border return time      $\triangleright$ Minimum iterations for $P_0$ to return to the left side
9:    $P_n \leftarrow \boldsymbol{A}_{\mathcal{R}}^n P_0 + (\boldsymbol{A}_{\mathcal{R}} - \boldsymbol{I})^{-1}(\boldsymbol{A}_{\mathcal{R}}^n - \boldsymbol{I})\boldsymbol{h}$     $\triangleright$ Compute $P_n$ directly from $P_0$
10:   $P_{n+1} \leftarrow$ Next iteration         $\triangleright$ Check for homoclinic intersection
11:   **if** $\mathsf{L}(x_n, y_n) \cdot \mathsf{L}(x_{n+1}, y_{n+1}) < 0$ **then**
12:      **Return** "Homoclinic intersection exists"
13:   **else**
14:      **Return** "No homoclinic intersection"
15:   **end if**
16: **end procedure**

---

*Remark* H.2. A similar algorithm can be devised for the fixed point $\mathcal{O}_{\mathcal{R}}^* \in \mathcal{R}$.

*Remark* H.3. Suppose that the first iteration of $P_0$ is on the left hand side ($P_1 \in \mathcal{L}$), but moves to the right side of the border after some iterations, i.e., there is some $k_0^* > 1$ such that $P_{k_0^*} \in \mathcal{R}$. Then a similar procedure could be applied for $P_{k_0^*}$ to find homoclinic intersections. In that case we can obtain the border return time needed for $P_{k_0^*}$ to return to the left hand side again. Using Proposition D.2, the iterations of $P_{k_0^*} = (x_0^*, y_0^*)^{\mathsf{T}}$ can be computed as

$$
P_{k_n^*} = \begin{pmatrix} x_n^* \\ y_n^* \end{pmatrix} = \begin{pmatrix} A_{n+1} - dA_n & cA_n \\ b_r A_n & dA_n - \mathcal{D}_{\mathcal{R}} A_{n-1} \end{pmatrix} \begin{pmatrix} x_0^* \\ y_0^* \end{pmatrix} + \frac{-1}{\mathcal{P}_{\mathcal{R}}(1)} \times
$$

$$\begin{pmatrix} 1-d & c \\ b_r & 1-a_r \end{pmatrix} \begin{pmatrix} A_{n+1} - dA_n - 1 & c\,A_n \\ b_r A_n & dA_n - \mathcal{D}_{\mathcal{R}} A_{n-1} - 1 \end{pmatrix} \begin{pmatrix} h_1 \\ h_2 \end{pmatrix}$$

$$= \begin{pmatrix} \left(A_{n+1} - d\,A_n\right)x_0^* + cA_n\,y_0^* - \Theta \\[2mm] b_r A_n x_0^* + \left(dA_n - \mathcal{D}_{\mathcal{R}} A_{n-1}\right)y_0^* - \frac{b_r\left[A_n(1-\Gamma_{\mathcal{R}}) + A_{n+1} - 1\right]h_1 + b^* h_2}{\mathcal{P}_{\mathcal{R}}(1)} \end{pmatrix} \tag{64}$$

where $\Theta = \dfrac{\left((1-d)\left[A_{n+1} - dA_n - 1\right] + cb_r A_n\right)h_1 + c\left[A_n - \mathcal{D}_{\mathcal{R}} A_{n-1} - 1\right]h_2}{\mathcal{P}_{\mathcal{R}}(1)}$. Investigating the signs of $\mathsf{L}(x_n^*, y_n^*) \cdot \mathsf{L}(x_{n+1}^*, y_{n+1}^*)$ we can then check for the existence of homoclinic intersections.

# I  Systems

## I.1  Duffing system

The *Duffing system* is a classical nonlinear dynamical system that models a damped and periodically driven oscillator with a nonlinear restoring force. Originally introduced by Georg Duffing [14], the system is described by a second-order differential equation of the form

$$\ddot{x} + \delta \dot{x} + \alpha x + \beta x^3 = \gamma \cos(\omega t),$$

where $x$ represents the displacement, $\delta$ is a damping coefficient, $\alpha$ and $\beta$ determine the linear and nonlinear stiffness respectively, and $\gamma \cos(\omega t)$ is an external periodic forcing term. The Duffing system exhibits rich dynamical behavior, including periodic, quasi-periodic, and chaotic responses, making it a prototypical example in the study of nonlinear and chaotic dynamics.

## I.2  Lorenz63

The *Lorenz63 system* [50] is a continuous-time dynamical system originally developed to model atmospheric convection. It describes the evolution of three state variables governed by the nonlinear differential equations

$$\frac{dx_1}{dt} = \sigma(x_2 - x_1),$$
$$\frac{dx_2}{dt} = x_1(\rho - x_3) - x_2,$$
$$\frac{dx_3}{dt} = x_1 x_2 - \beta x_3,$$

where $x_1$, $x_2$, and $x_3$ denote, respectively, the convection rate, horizontal temperature difference, and vertical temperature difference. The parameters $\sigma$, $\rho$, and $\beta$ correspond to physical constants related to the Prandtl number, Rayleigh number, and system geometry.

For specific values, e.g. $\sigma = 10$, $\rho = 28$, and $\beta = \frac{8}{3}$, the system exhibits chaotic dynamics. These settings give rise to the well-known "butterfly attractor," a prime example of deterministic chaos in low-dimensional systems.

# J  Evaluation Metrics

## J.1  Geometrical Measure $D_{\text{stsp}}$

Given probability distributions $p(\boldsymbol{x})$ (estimated from ground truth trajectories) and $q(\boldsymbol{x})$ (estimated from model-generated trajectories), $D_{\text{stsp}}$ is defined as the Kullback-Leibler (KL) divergence

$$D_{\text{stsp}} := D_{\text{KL}}(p(\boldsymbol{x}) \parallel q(\boldsymbol{x})) = \int_{\mathbf{x} \in \mathbb{R}^N} p(\boldsymbol{x}) \log \frac{p(\boldsymbol{x})}{q(\boldsymbol{x})} \, d\boldsymbol{x}. \tag{58}$$

For low-dimensional observation spaces, $p(\boldsymbol{x})$ and $q(\boldsymbol{x})$ can be estimated using binning [44, 6]. The KL divergence is approximated as

$$D_{\text{stsp}} = D_{\text{KL}}(\hat{p}(\boldsymbol{x}) \parallel \hat{q}(\boldsymbol{x})) \approx \sum_{k=1}^{K} \hat{p}_k(\boldsymbol{x}) \log \frac{\hat{p}_k(\boldsymbol{x})}{\hat{q}_k(\boldsymbol{x})}. \tag{59}$$

Here, $K = m^N$ is the total number of bins, with $m$ bins per dimension. $\hat{p}_k(\boldsymbol{x})$ and $\hat{q}_k(\boldsymbol{x})$ are the normalized counts in bin $k$ for ground truth and model-generated orbits, respectively.

In high-dimensional settings, Gaussian Mixture Models (GMMs), placed along the trajectories, are used [6]. This results in approximate probability distributions

$$\hat{p}(\boldsymbol{x}) = \frac{1}{T'} \sum_{t=1}^{T'} \mathcal{N}(\boldsymbol{x}; \{\boldsymbol{x}_t\}, \Sigma), \quad \hat{q}(\boldsymbol{x}) = \frac{1}{T'} \sum_{t=1}^{T'} \mathcal{N}(\boldsymbol{x}; \{\hat{\boldsymbol{x}}_t\}, \Sigma)$$

where the covariance matrix is given by $\Sigma = \sigma^2 \mathbf{1}_{N \times N}$ ($\sigma$ is a hyperparameter), and $\{\boldsymbol{x}_t\}, \{\hat{\boldsymbol{x}}_t\}$ are samples from the true and generated orbits of length $T'$.

Using the Monte Carlo approximation from Hershey and Olsen [34], the KL divergence in this case is estimated by

$$D_{\text{stsp}} = D_{\text{KL}}(\hat{p}(\boldsymbol{x}) \parallel \hat{q}(\boldsymbol{x})) \approx \frac{1}{n} \sum_{i=1}^{n} \log \frac{\hat{p}(\boldsymbol{x}^{(i)})}{\hat{q}(\boldsymbol{x}^{(i)})} \tag{60}$$

with $n$ Monte Carlo samples $\{\boldsymbol{x}^{(i)}\}$ randomly drawn from the GMM $\hat{p}(\boldsymbol{x})$ that represents the real data distribution.

## J.2  Prediction Error PE

The $n$-step prediction error is defined as the mean squared error between ground truth data $\{\boldsymbol{x}_t\}$ and $n$-step ahead predictions of the model $\{\hat{\boldsymbol{x}}_t\}$, i.e.

$$\text{PE}(n) = \frac{1}{N(T-n)} \sum_{t=1}^{T-n} \|\boldsymbol{x}_{t+n} - \hat{\boldsymbol{x}}_{t+n}\|_2^2. \tag{65}$$

 # K    Additional figures

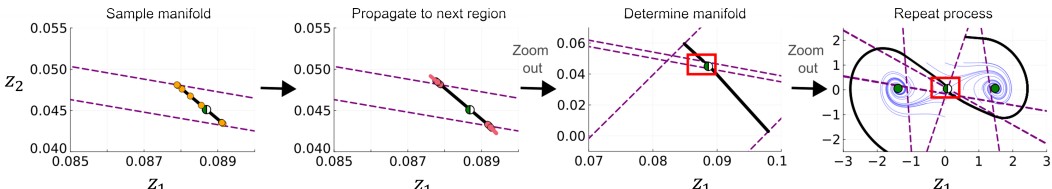

Figure 18: Illustration of the iterative procedure for computing stable manifolds with subregion boundaries of the shPLRNN ($M = 2$, $H = 10$) model in purple (dashed) Step 1: The stable manifold (black) is initialized using the stable eigenvector at the saddle point (half-green), and sample points (orange) are placed along it. Step 2: These points are propagated until they reach a new linear subregion, where the flow field is evaluated. Step 3: The updated manifold is given by the first principal component of points and the flow. Step 4: Repeating this process iteratively reconstructs the full global structure of the stable manifold (black), overlaid with the underlying GT flow field (blue)

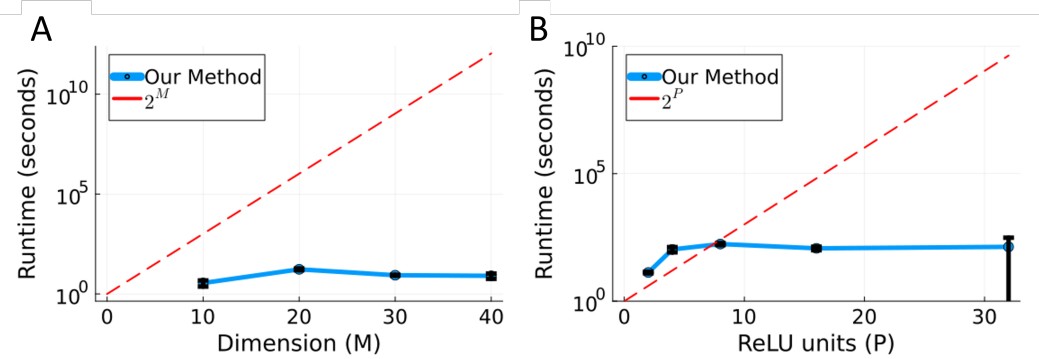

Figure 19: Algorithm runtime for determining the stable manifold of a saddle point averaged across 5 runs (error bars: standard deviation). A: For a *constant* number of linear subregions $2^P$, Algo. 1's runtime hardly increases as a function of model size $M$ for an ALRNN ($P = 2$), confirming it is not significantly affected by the number of model parameters per se. B: For a *constant* model size $M$, Algo. 1's runtime increases much slower than $2^P$ for an ALRNN ($M = 40$) when the manifold construction is restricted to the set of linear subregions explored by the data. All models were trained on the Duffing system for within-comparability, but we emphasize that the scaling may strongly depend on the system's actual dynamics and topological structure, such that general statements regarding scaling are therefore difficult. Runtime was determined on an Intel Core i5-1240P.

