# OpenReview forum: "Detecting Invariant Manifolds in ReLU-Based RNNs"
_NeurIPS.cc/2025/Conference — Submitted to NeurIPS 2025_

### Official Review · Reviewer_G6w7 · 2025-06-30

**Clarity:** 3
**Significance:** 2
**Originality:** 3
**Rating:** 4
**Confidence:** 3

**Summary:**

The authors present a novel algorithm for computing stable and unstable manifolds in piecewise-linear recurrent neural networks (PLRNNs) with ReLU activations. By exploiting the piecewise-linear structure of PLRNNs, the algorithm constructs stable and unstable manifolds using eigenvalues and eigenvectors. It is also capable of detecting homoclinic and heteroclinic orbits. The paper also analyzes both synthetic systems and real neural data.

**Questions:**

1. It might be better to reorder citation in the paper.
2. How sensitive is the algorithm to slight perturbation of invertibility?
3. Could this method be extended to non-piecewise-linear activations?
4. Are there specific types of dynamics where the algorithm fails?

**Ethical Concerns:**

["NO or VERY MINOR ethics concerns only"]

**Final Justification:**

All of my concerns have been addressed by the authors. I have no further questions, and I now hold a positive evaluation of this paper.

**Limitations:**

yes

**Quality:**

3

**Strengths And Weaknesses:**

This is the first algorithm which can detect stable/unstable manifolds in ReLU-based RNNs. The authors conduct a comprehensive analysis with both simulated data and neural data.  However, one of the main challenges in analyzing RNNs lies in their nonlinearities, such as ‘tanh’, however, the authors only focused on Relu-based RNN (PLRNN in the paper), which are piecewise-linear and allow fixed points to be more readily identified via eigenvalues and eigenvectors.
The method assumes invertibility of the RNN map, which may not hold for all trained networks. They mitigate this via a regularization term, but it still limits generality. (As acknowledged in the paper.)
Therefore, my score will be borderline.

---

> ### Author Rebuttal · Authors · 2025-07-30
>
> We thank the referee for the thorough evaluation of our work and the helpful feedback provided.
>
> **Weaknesses**
>
> **W1 (focus on piecewise linear systems):** While it is true that our algorithm exploits the particular properties of piecewise linear (PL), ReLU-based, RNNs, we would like to point out that this nevertheless encompasses a broad  class of current SOTA dynamical systems reconstruction (DSR) models, such as almost-linear (AL) RNNs [1], shallow PLRNNs [2], dendritic PLRNNs [3], or recurrent switching linear DS [4, 5], and that, more generally, PL models have a famous history in the mathematics of DS [6,7] and in engineering [8,9] for their tractability properties. We believe this is a huge advantage of using PL models in this field of DSR, for which our algorithm is intended.
>
> This being said, we looked into the possibility of extending our algorithm to RNNs with smooth activation functions such as tanh, and provide a modified algorithm for such cases below which rests on local linearizations. The suggested algorithm proceeds by locally linearizing the manifold from a set of points by PCA. A set of support points is then propagated forward (for unstable manifolds) or backward (for stable manifolds) via the RNN flow map (for which we can encourage invertibility via regularization). As long as these points remain well-approximated by the current locally linear manifold as defined by PCA, we continue propagating within the same linear patch. When the current linear model no longer provides an adequate approximation, as determined by a threshold on the reconstruction error (see below), a new locally linear manifold is constructed via PCA on these propagated points.
>
> While this procedure is no longer exact (as for PL models) and computationally more demanding (since we cannot determine the manifold in an entire linear subregion but only locally within a patch), it still retains some of the scalability and interpretability of the original approach, without requiring a PL form of the RNN. To determine when the locally linear approximation becomes inaccurate, we compute the PCA reconstruction error on the newly propagated points $S\_\\text{new} = \\{ s\_i\\}\_{i=1}\^N$ assuming the current local patch $Q\_k$ (see [10]). If $\\text{err}\_\\text{rec} > \\epsilon\_\\text{rec}$, we start a new manifold patch by fitting a new local PCA model. This test detects whether the manifold curvature significantly increases and the support points start deviating from the current locally linear structure.
>
> We tested this idea on a vanilla RNN with tanh activation trained on the Duffing system which has a curved stable manifold (cf. Fig. 4a), and observed that the reconstructed stable manifold of the saddle agreed very well with the model’s vector field (as in Fig. 4a). While unfortunately the current rebuttal policies don’t allow for including additional graphs, we will include these results in the revision.
>
> **Algorithm 1:** *Manifold Construction for Smooth RNNs*
>
> **Input:**
> &nbsp;&nbsp;&nbsp;&nbsp;$N_\{max\}$: Maximum number of iterations
> &nbsp;&nbsp;&nbsp;&nbsp;$P$: periodic point
> &nbsp;&nbsp;&nbsp;&nbsp;$σ$: Index indicating whether the stable or unstable manifold is to be computed
> &nbsp;&nbsp;&nbsp;&nbsp;$E^\sigma$: Stable/unstable eigenvectors
> &nbsp;&nbsp;&nbsp;&nbsp;$\lambda^\sigma$: Stable/unstable eigenvalues
> &nbsp;&nbsp;&nbsp;&nbsp;$\epsilon$: Threshold for deciding when to create a new patch
>
> **Output:**
> &nbsp;&nbsp;&nbsp;&nbsp;$\\{S^\sigma_k, Q^\sigma_k\\}^{k_{max}}_{k=0}$: Sampled points and local manifold models
>
> ---
>
> 1. $k \leftarrow 0$ ▷ Index of current manifold patch
> 2. $Q^\sigma_k \leftarrow \text{GetManifold}(P, E^\sigma, \lambda^\sigma)$ ▷ Initialize linear manifold
> 3. $S^\sigma_k \leftarrow \text{SamplePoints}(Q^\sigma_k)$ ▷ Sample initial points on manifold
> 4. **for** $n = 0 : N_{\text{max}}$ **do**
>    - $S^\sigma_{\text{new}} \leftarrow \text{Propagate}(S^\sigma_k, \sigma)$ ▷ Propagate forward/backward depending on $\sigma$
>    - $err_\{rec\} \leftarrow ReconstructionError(S^\sigma_\{new\}, Q^\sigma_k)$ ▷ PCA reconstruction error
>    - **if** $\text{err}_{\text{rec}} > \varepsilon$ **then**
>      - $k \leftarrow k + 1$
>      - $Q^\sigma_k \leftarrow \text{PCA}(S^\sigma_{\text{new}})$ ▷ New local manifold patch
>      - $S^\sigma_k \leftarrow \text{SamplePoints}(Q^\sigma_k)$ ▷ Resample within new patch
>    - **end if**
> 5. **end for**
> 6. **return** $\\{S^\sigma_k, Q^\sigma_k\\}^{k_{max}}_{k=0}$ ▷ Return patches and samples
>
> **W2 (invertibility of RNN map):** The invertibility requirement is only a *very mild* constraint in practice, as almost all systems of interest which can be described by differential equations will be invertible almost everywhere. This is a direct consequence of the Picard-Lindelöf theorem guaranteeing the uniqueness of solutions for initial value problems for sets of continuously differentiable differential equations [11]. For such systems, the flow operator (which we are attempting to approximate by the RNN [12]) will be a diffeomorphism (hence invertible). Even more generally, when the vector field is continuous (but not differentiable) and satisfies a local Lipschitz condition, the flow is still a homeomorphism, thus invertible (via time-reversal) with a continuous inverse [13, 14, 15]. For continuous systems with attractors, the set of initial conditions for which the flow is non-invertible is negligible and will generally have measure zero (e.g. on point attractors). Thus, in practice this is not really a relevant limitation (as evidenced by the different examples in sect. 4), and we may have actually been too cautious in the statements in our paper.
>
> It might also be important to note that by ‘invertibility’ here we do not mean invertibility of the model’s Jacobian (transition matrix), but we mean that there $\exists ! \\, \\boldsymbol{D}\_{t-1}$ for which $F(F^{-1}(z_t,\\boldsymbol{D}\_{t-1}))=z_t$ (i.e., such that the mapping $F$ is bijective *across boundaries*). That is, even at this level the condition is actually much less restrictive than the referee may have assumed.
>
> **Questions**
>
> **Q1)** Thanks for pointing out, we will make sure that citations will be monotonically ordered in the revision.
>
> **Q2)** Please see reply to W2 above. Further to this, invertibility does not strictly need to hold everywhere, since a local violation of the invertibility condition only means that there may be no path from any given subregion into a current one (see def. above in W2), i.e. that a current candidate subregion will have no preimage in any other region, which in turn implies the manifold simply does not extend into that subregion (as we now also numerically confirmed, with a strong linear relation, $R^2=0.985$, between the number of subregions visited and the number of subregions for which invertibility holds).
>
> **Q3)** Yes, please see reply to W1 above.
>
> **Q4)** The algorithm can correctly identify un-/stable manifolds of all fixed and cyclic points, there are no limitations in this respect. However, as mentioned in our Limitations sect., although the algorithm can strictly determine the presence of chaos, it will struggle with reconstructing the full extent of un-/stable manifolds of chaotic sets because of their fractal geometry. This, we hope, we already made explicit in the Limitations.
>
> **References:**
>
> [1] Brenner et al. Almost‑Linear Recurrent Neural Networks Yield Highly Interpretable Symbolic Codes in Dynamical Systems Reconstruction. In NeurIPS 2024
>
> [2] Hess et al. In Proceedings of the 40th International Conference on Machine Learning, pages 13017–13049. PMLR, July 2023.
>
> [3] Brenner et al.  Tractable Dendritic RNNs for Reconstructing Nonlinear Dynamical Systems. In Proceedings of the 39th International Conference on Machine Learning, pages 2292–2320. PMLR, June 2022.
>
> [4] Linderman et al. Bayesian Learning and Inference in Recurrent Switching Linear Dynamical Systems. In Proceedings of the 20th International Conference on Artificial Intelligence and Statistics, pages 914–922. PMLR, April 2017.
>
> [5] Alameda-Pineda et al. Variational Inference and Learning of Piecewise Linear Dynamical Systems. IEEE Transactions on Neural Networks and Learning Systems, 33(8):3753–3764, August 2022
>
> [6] Alligood et al. Chaos: An Introduction to Dynamical Systems. Textbooks in Mathematical Sciences. Springer, 1996.
>
> [7] Lind et al. An Introduction to Symbolic Dynamics and Coding. Cambridge University Press, Cambridge, 1995.
>
> [8] Carmona et al. On simplifying and classifying piecewise-linear systems. IEEE Transactions on Circuits and Systems I: Fundamental Theory and Applications, 49(5):609–620, 2002.
>
> [9] Sontag. Nonlinear regulation: The piecewise linear approach. IEEE Transactions on Automatic Control, 26(2):346–358, April 1981
>
> [10] Reiß and Wahl. Nonasymptotic upper bounds for the reconstruction error of PCA. The Annals of Statistics 48, no. 2 (2020): 1098-1123.
>
> [11] Perko (2001). Differential Equations and Dynamical Systems. 3rd ed. Texts in Applied Mathematics 7. New York: Springer-Verlag.
>
> [12] Goering et al. (2024). Out‑of‑Domain Generalization in Dynamical Systems Reconstruction. Proceedings of the 41st International Conference on Machine Learning (ICML), PMLR 235, 16071–16114.
>
> [13] Hirsch et al. (2013). Differential Equations, Dynamical Systems, and an Introduction to Chaos (3rd ed.). Academic Press.
>
> [14] Coddington and Levinson (1955). Theory of Ordinary Differential Equations. McGraw-Hill.
>
> [15] Arnold (1992). Ordinary Differential Equations (3rd ed., translated by R. Cooke). Springer.

---

> > ### Comment · Reviewer_G6w7 · 2025-08-01
> >
> > I thank the authors for their response. All of my concerns have been addressed. I have no further questions and will raise my score to a positive evaluation.

---

> > > ### Author Response · Authors · 2025-08-01
> > >
> > > Thank you once again for your very constructive and helpful review, we are happy to hear we could address all your points!

---

### Official Review · Reviewer_JfiB · 2025-06-30

**Clarity:** 3
**Significance:** 4
**Originality:** 4
**Rating:** 5
**Confidence:** 4

**Summary:**

The authors propose a well-motivated heuristic algorithm for finding the stable and unstable manifolds of piecewise linear RNNs. Finding these is useful for illuminating the behavior of dynamical systems, as it can be used to illustrate e.g., the different basins of attractions for different choices in a decision making model. The authors also nicely illustrate the emergence of chaos without relying on mean-field theories / infinitely large networks.

**Questions:**

Could you explain why PCA is used? Is it simply more convenient to work with an orthonormal basis?

I also don’t understand the use of kernel-PCA, could you elaborate on this? If it is just to get a real basis PCA([Re(vecs), Im(vecs)]) should be sufficient, no?

**Ethical Concerns:**

["NO or VERY MINOR ethics concerns only"]

**Final Justification:**

This paper demonstrates a promising approach to an important open question (finding boundaries between attractors), for which there are as far as I know little good alternatives available. My questions with regards to what is heuristic and what is exact are sufficiently clarified.

**Limitations:**

yes

**Paper Formatting Concerns:**

No concerns

**Quality:**

3

**Strengths And Weaknesses:**

**Strengths**

This paper tackles an important open question in the field. The contributed method will be a useful tool to have available when reverse engineering dynamical systems. The paper is original, as there are not many alternatives for obtaining the manifolds. The empirical results are encouraging, and include a nice real world application to a single-cell recording.

**Weaknesses**

1. I would have liked a bit more concrete discussion on which parts of the algorithm are heuristic and which are exact. And when they are heuristic, how do potential hyper parameter choices affect the outcome, and how are they picked? I will give some concrete examples here:
- Use of approximate algorithm for finding the fixed points: Is there any concern in case one doesn't find all fixed points?
- Sampling seed points: how many are sampled? How are they sampled? It doesn’t seem trivial (or even always possible) to me to e.g., uniformly sample within one region of linear dynamics.
- Propagating seed points. I think one is not always sure we reach all reachable manifolds from a given sub-region? (E.g., if we undersample seed points, or all seed points / most of the space ends up converging in one particular direction).
- The fallback algorithm 3 still doesn’t have any guarantees of finding the right subspace, does it?

2. From results shown it is unclear to me that the regularisation (Eq. 6) actually reaches the desired effect. Besides the effect on training, does it actually lead to networks that are more likely to be invertible? Is there some way to verify this, maybe by comparing the average condition numbers of the to be inverted matrices when running your manifold construction algorithm on a model trained with and without the regulariser?


Note that while I do think some clarity on these will make the paper stronger, both the good empirical results, and the originality, make this a paper a valuable contribution!

------

Minor:
- Line 45: Maybe it can be useful to name the term separatrix (formed by the stable manifolds)?
- Line 113: In Pals et al., NeurIPS, 2024, it has been shown that the amount of regions in shPLRNNs is $\sum_m^M {H \choose m}$ ($\leq$ the stated $2^H$). This means your lorenz-63 model has only ~1300 instead of over a million regions, might be useful.
- Line 208: I think a regularisation does not *enforce* (it is a soft constraint).

---

> ### Author Rebuttal · Authors · 2025-07-30
>
> We thank the referee very much for the supportive, detailed, and thorough review! The referee brought up a couple of important points that need more attention and clarification.
>
> **Weaknesses**
>
> **W1 (heuristic vs. exact parts of algo):** Thank you for bringing this up. In brief, locally in the linear subregion, in which the fixed or cyclic point resides, the computation of the manifolds is exact and does not depend on any hyperparameters, but to determine how the manifolds extend into other linear subregions some numerical steps and hyperparameters are involved. These are all related to the sampling of points with the forward or backward RNN map, and include 1) the total number of points sampled (mainly to ensure all connected linear subregions are reached), 2) the perturbation size chosen, and 3) the weighing of fast vs. slow eigendirections in sampling to avoid numerical issues in cases of strong anisotropy in the vector field. We will make sure to clarify this in our revision.
>
> On your detailed points:
>
> 1) Please note that the algorithm for finding fixed and cyclic points (SCYFI [1]) itself strictly is not part of the present algorithm. It is approximate in the sense that not all fixed points may be found, but the fixed points themselves will be exact. While SCYFI has been shown to have generally good convergence properties [1], if not all fixed points are found this simply means that we cannot determine all of the un-/stable manifolds. Whether this is of concern we would argue depends more on the specific application case, as often obtaining a local picture of the state space may be sufficient. But this may be important to point out as a potential limitation, and so we will add this to our discussion.
>
> 2) Generally speaking, we need to sample as many seed points as necessary to uniquely position the $K$-dimensional manifold ($K \leq M$) within each subregion, which means that sometimes $K+1$ points per subregion may be enough. Please note that points don’t need to be sampled *uniformly within a subregion*, this may be a misunderstanding, but are randomly sampled (with intervals drawn from a uniform distribution) *along the local stable or unstable manifolds*. And only as many points need to be sampled as strictly necessary to uniquely position that segment of the manifold, e.g. 2 points will be enough to specify a line segment within one subregion *in the absence of numerical issues*. Thus, we do not need to cover the whole manifold (or subspace) either. To avoid numerical issues, however, as described in  Sec. 3.3, the density of sampled points is further locally adapted to account for strong anisotropy or curvature of the vector field (as inferred from the eigenvalues, see Sec. 3.3). We will clarify this important point in our revision.
>
> 3) This is, in principle, a correct observation, we cannot strictly guarantee all *linear subregions* are reached (once a subregion is reached, however, we can guarantee reconstruction of the full manifold within that subregion). Here the number of sampled points becomes kind of a hyperparameter – if the number of detected continuations of the manifold into other subregions does not increase anymore for some value $N_\\text{samp}$ of sampled points, we can be reasonably sure we have recovered the full extent of the manifold. In the case of the Duffing system, for instance, we were able to reconstruct the manifold with just 4 sample points per region. Again, we agree this needs clarification.
>
> 4) Here the same argument applies as above, i.e. we could in principle examine the dependence on $N_\\text{samp}$ (the number of sample points) to determine the point where the number of accessed subregions plateaus.
>
> **W2 (effect of regularisation):** Yes, the regularisation does produce the desired effect, as we tried to demonstrate in Fig. 2A (top-right). But perhaps we failed to make sufficiently clear that by ‘invertibility’ we do not mean invertibility of the model’s Jacobian (transition matrix), but we mean that there $\exists !\\, \\boldsymbol{D}\_{t-1}$ for which $F(F^{-1}(z_t,\\boldsymbol{D}\_{t-1}))=z_t$ (i.e., such that the mapping $F$ is bijective across boundaries). For this we need the determinants of all Jacobians to have the same sign [2], as encouraged by Eq. 6. We need this condition for backtracking the stable manifolds *across linear subregions*. We will make this point very clear in our revision.
>
> **Minor points**
>
> **M1)** Agreed, we will do so.
>
> **M2)** Yes, thanks for pointing out, we will amend this and cite the source. The reason why we provided the “upper bound” on this number is that it is unclear a priori, without further investigation, which of the $2^H$ “hidden” regions will “collapse” into the same region in $M$-dimensional latent space, and that it is computationally too expensive to track this throughout training.
>
> **M3)** Yes true, we will change our wording to “encourage”.
>
> **Questions**
>
> **Q1 (why PCA):** PCA in this context is simply a convenient way to precisely locate the manifold, it is not used in the conventional sense of finding variance-maximizing directions, but to determine the precise orientation of the manifold. It is necessary because the number of stable and unstable eigendirections may change as one hops between linear subregions, so we cannot simply use all the eigenvectors, but need to detect the subspace harboring the manifold. Again, this needs clarification in the manuscript.
>
> **Q2 (why kernel-PCA):** Note that these manifolds may also be curved (see Eq. 4 and Fig. 4A,C), and kernel-PCA provides us with a means to exactly anchor the manifold in this case by capturing its curvature (with the right kernel, e.g. exponential-trigonometric [3]). However, practically speaking the referee is right, local PCA often provides a good approximation (depending on how well the assumption of local linearity holds).
>
> **References:**
>
> [1] Eisenmann et al. Bifurcations and loss jumps in RNN training. Advances in Neural Information Processing Systems, 36, 2024
>
> [2] Fujisawa et al. A sparse matrix method for analysis of piecewise linear resistive networks. IEEE Transactions on Circuit Theory, 19(6):571–584, 1972.
>
> [3] Kosikova et al. Bayesian structural identification using Gaussian process discrepancy models. Computer Methods in Applied Mechanics and Engineering, 2023, 417. Jg., S. 116357.

---

> ### Comment · Reviewer_JfiB · 2025-08-01
>
> I thank the authors for their in-depth answer and trust that they will clarify the sampling of points and their hyperparameter settings (number chosen, perturbation size and weighting) in the final manuscript. I am also intrigued by the expansion to $tanh$ networks, prompted by comments of other reviewers, which seems an *additional* valuable contribution.
>
> All my questions are clarified, except one - I am still slightly confused by the use of (kernel) PCA.
> >local PCA often provides a good approximation
>
> Since dynamics within one subregion are linear, isn't the space spanned by the local eigenvector / PC basis (restricted to their own subregion) always enough?
>
> Or is the goal of the PCA to have one concise representation of the *global* manifold? What exactly is it that then goes in to the PCA, points sampled along all the local manifolds?

---

> > ### Author Response · Authors · 2025-08-01
> >
> > We completely understand the referee’s confusion, as exactly this point initially caused some discussion among ourselves as well!
> >
> > Regarding the subregion in which the fixed or cyclic point resides (call it $S_0$), the referee is completely right, the manifold is spanned by the un-/stable eigenvectors. However, as soon as we start propagating into a neighboring region, say $S_1$, we are no longer talking about the un-/stable eigendirections within $S_1$, but about the sets in $S_1$ that have their image on the stable manifold in $S_0$ or their preimage on the unstable manifold in $S_0$ (which are not identical to the un-/stable eigendirections in $S_1$). We know, however, that a) the dimensionality of the manifold itself must stay the same, and b) the general solution is given by Eq. 4 and may involve curvature (if eigenvalues have imaginary parts). If the imaginary parts are zero and there are no eigenvalues with multiplicity $>1$, we can use PCA to find the right orientation of the new segment of un-/stable manifold in $S_1$ based on the sample points (which, by construction, lie on the resp. manifold). If the imaginary parts are non-zero, the manifold is curved (Eq. 4) and we can use kernel-PCA to position it.

---

> > > ### Comment · Reviewer_JfiB · 2025-08-01
> > >
> > > > we are no longer talking about the un-/stable eigendirections within $S_1$, but about the sets in $S_1$ that have their image on the stable manifold in $S_0$ or their preimage on the unstable manifold in $S_0$ (which are not identical to the un-/stable eigendirections in $S_1$).
> > >
> > > Thanks, that clarifies it!
> > >
> > > I retain my (very) positive score.

---

> > > > ### Author Response · Authors · 2025-08-01
> > > >
> > > > Thank you very much for your detailed and helpful questions, and for your supporting assessment, we will make sure to include these clarifications in our revision!

---

### Official Review · Reviewer_nWbu · 2025-07-03

**Clarity:** 3
**Significance:** 2
**Originality:** 3
**Rating:** 4
**Confidence:** 1

**Summary:**

This paper presents a novel semi-analytical method for identifying invariant manifolds (stable/unstable manifolds of saddle points) in piecewise-linear RNNs. By exploiting the network’s piecewise-linear structure, the method computes the local manifold in the linear region of a saddle via eigen-decomposition, then extends it globally across regions by sampling and iteratively propagating points. The algorithm thus delineates basin boundaries revealing multistability and detects homoclinic/heteroclinic intersections revealing chaos, as demonstrated on toy maps, classical dynamical systems, and neural data.

**Questions:**

Please refer to the above-mentioned weaknesses.

**Ethical Concerns:**

["NO or VERY MINOR ethics concerns only"]

**Final Justification:**

The authors have addressed all my questions, and I am inclined to maintain the positive score I gave. In their response to Weakness 1, the authors stated, "we already created one, but unfortunately are not allowed to provide figures with the revision." You may include the figures of this example in your reply.

**Limitations:**

Yes

**Quality:**

3

**Strengths And Weaknesses:**

# Strength:
1. This work introduces the ​​first dedicated algorithm​​ for computing invariant manifolds in ReLU-based PLRNNs, leveraging their piecewise-linear structure to bypass traditional limitations of numerical continuation methods. By analytically deriving manifolds within linear subregions and propagating them iteratively, the approach achieves polynomial-time scalability beyond the 5D barrier of prior techniques, filling a critical gap in high-dimensional dynamical systems analysis.
2. The study establishes rigorous mathematical foundations - formally defining invariant manifolds, basins of attraction, and homoclinic chaos - while deriving closed-form solutions for dynamics in affine subregions. The novel invertibility regularization ensures theoretical consistency with minimal computational overhead, demonstrating principled innovation throughout.
3. Validated across synthetic and real-world benchmarks, the algorithm robustly handles high-dimensional models with sublinear scaling in subregion sampling. Its efficiency enables previously infeasible analyses, such as identifying attractor basins in decision-making tasks or chaos signatures in electrophysiological data, bridging theoretical insights with empirical applications.
4. Unifying traditionally disjoint analyses, the framework simultaneously delineates basins of attraction, detects chaotic regimes via homoclinic intersections, and extracts mechanistic principles. This integration provides a versatile toolkit for dissecting RNN dynamics, from memory states to edge-of-chaos computational properties.

Weakness:
1. Medical applications require guarantees that chaotic dynamics won't amplify adversarial perturbations, yet no experiments probe this vulnerability. The discussion in Sec. Limitations ignore adversarial threats in high-stakes settings, focusing solely on invertibility constraints.
2. The paper exhibits ​​significant bias in application scope validation​​, disproportionately focusing on cortical neuron dynamics and classical physical systems, while neglecting critical and general domains explicitly mentioned in the introduction. Claimed applications in "time series prediction" in the abstract remain untested for ​​sensor networks or fault detection​​.
3. The algorithm's theoretical O(2^P) complexity risks intractability in high-dimensional industrial systems. Validation remains narrowly focused on low-dimensional models and neuroscience tasks, omitting critical industrial scenarios. It lacks the stress-testing on high-noise/long-horizon data and real-time control benchmarks.

---

> ### Author Rebuttal · Authors · 2025-07-30
>
> We thank the referee for taking the time and effort to scrutinize our work. Although the referee gave a confidence score of just 1, the Summary and Strengths describe the essence of our algorithm very accurately!
>
> **Weaknesses**
>
> **W1 (adversarial perturbations):** The risk that a dynamical system is derailed through perturbations, either by its sensitivity due to the presence of chaos or due to tipping across basin boundaries, is indeed a serious issue in medical and other high-stake settings, we fully agree! However, we think it is *precisely a strength of our algorithm that it allows to assess such risks* based on data-trained RNNs. Note that our paper is not dealing with modeling dynamical systems in itself, but rather with the *analysis of trained models* which may be intended for medical use for instance. We can directly use our algorithm to assess whether a model is vulnerable to perturbations or adversarial threats by analyzing the structure of its state space. Since our algorithm computes the exact location and geometry of basin boundaries in the latent state space, we can determine the minimal distance of an attractor or any given state to the nearest basin boundary, which directly corresponds to the smallest perturbation that could push the system into a different dynamical regime (noise-induced tipping [1]).
>
> For instance, assume we trained a model on human electrocardiogram (ECG) data and found bistability between a healthy activity regime and a pathological one. We can then use our algorithm to compute the basin boundary between these two regimes and determine the distance between activity on the healthy cardiac cycle and the boundary towards the pathological state. If this ‘safety margin’, the minimal distance to the basin boundary, is small, the system is inherently vulnerable to fluctuations induced by noise or various physiological stressors. Moreover, by combining this with local eigenvalue spectra, we can identify directions in state space along which perturbations would be most strongly amplified (e.g., along unstable manifolds), potentially providing valuable information for physicians. Likewise, we could use our algorithm to identify chaotic regimes in trained RNNs, and to suggest variations in parameters that may stabilize/ control the system.
>
> We will include this discussion of potential use cases of our algorithm in the Conclusions sect. of our paper, and will provide an illustrative example for such a configuration (we already created one, but unfortunately are not allowed to provide figures with the revision).
>
> **W2 (range of applications):** Note that our algorithm is designed to dissect the topological and geometrical structure of piecewise-linear RNNs, regardless of which type of data these have been trained on. It is completely agnostic to the source of the data used for training the RNN, and hence the mentioned limitation in our minds would apply more to actual RNN models or training algorithms but not so much to our algorithm for *analyzing* such models. Simply for illustrative purposes we have decided to pick examples from physics and neuroscience with which we ourselves are most familiar with, such that we are able to interpret the analysis results and adequately put them into the context of the resp. literature. But this doesn’t reflect a limitation of the algorithm itself.
>
> **W3 (theo. complexity/ industrial scenarios):** As above (cf. W1, W2), high-noise and long-horizon data are an issue of proper *model training*, not an issue of our model analysis algorithm which inherently doesn’t care about the amount or length of time series data available for training, or their noisiness. Our algorithm provides a method for dissecting the topological and geometrical structure of RNN state spaces, *regardless of what type of data the RNN previously has been trained on*! On the contrary, as discussed in W2 above, it could provide a means, for instance, to *analyze the vulnerability of the RNN or underlying system to noise*. This is a point and application we will make much clearer in our revision, as discussed in our replies above. Please also note that the model trained on real neurophysiological data (Fig. 4d) is indeed high-dimensional (the additional example we now created, see W1, is 60d). Also please note that in practice $P$ is usually rather small ($\leq 10$, see [2]) and $O(2^P)$ is only the worst case scenario (cf. [3]).
>
> **References:**
>
> [1] Ashwin et al. Tipping points in open systems: bifurcation, noise-induced and rate-dependent examples in the climate system. Phil. Trans. R. Soc. A 370, 1166-1184, 2012
>
> [2] Brenner et al. Almost‑Linear Recurrent Neural Networks Yield Highly Interpretable Symbolic Codes in Dynamical Systems Reconstruction. In NeurIPS 2024
>
> [3] Eisenmann et al. Bifurcations and loss jumps in RNN training. Advances in Neural Information Processing Systems, 36, 2024

---

> > ### Comment · Reviewer_nWbu · 2025-08-06
> >
> > In your response to W1, you stated, "we already created one, but unfortunately we are not allowed to provide figures with the revision." You may include the figures for this example in your official comment.

---

> > > ### Author Response · Authors · 2025-08-06
> > >
> > > Dear Referee,
> > >
> > > The rebuttal guidelines explicitly state the following (email sent on 7/27):
> > >
> > > "  2. Because of known concerns on identity leakage, we prohibit using any links in the rebuttal, including but not limited to anonymous or non-anonymous URL links, or updating your submitted github repository.
> > > We understand that 1) and 2) effectively prevent using image/video or other rich media as a communication protocol between reviewers and authors. We are sorry about that.
> > > We ask everyone’s help to respect the policies above. **For authors, please refrain from submitting images in your rebuttal with any “tricks”** (such as encoding your image in base64). We will allow reviewers/ACs/SACs to disregard those (and **even decrease their scores because of those**) when they spot the intent.
> > > For reviewers/ACs/SACs, please do not expect the authors to submit updated results with images or any type of rich media. "
> > >
> > > Our example is, however, similar in spirit to the bistable Duffing system in Fig. 4A (but much higher-dimensional), where perturbations could drive the system from one stable regime into another, for which we will illustrate how to compute the distance to the basin boundary (in principle, or additionally, one could also harvest the bistable cell example from Fig. 4D for this).
> > >
> > > Thank you once again for your feedback on our manuscript!

---

### Official Review · Reviewer_1awS · 2025-07-03

**Clarity:** 3
**Significance:** 3
**Originality:** 4
**Rating:** 4
**Confidence:** 3

**Summary:**

This paper introduced an algorithm to detect stable and unstable manifolds in ReLU-based piecewise-linear RNNs. By exploiting the network's piecewise-linear structure, the method iteratively constructs global invariant manifolds via local eigenvectors and adaptive sampling, tracing boundaries between basins of attraction. The authors also introduced an invertibility regularization to ensure the RNN map remains invertible, enabling robust manifold tracing even in high dimensions. Demonstrations on synthetic and real neural data show the method’s utility for characterizing multistability, detecting chaos, etc.

**Questions:**

As mentioned in the weaknesses, your algorithm relies on the RNN being piecewise linear in order to define and track linear subregions. Do you see any possible ways to approximate or extend your approach so that it could be applied to more general RNNs with smooth or non-piecewise-linear activation functions?

**Ethical Concerns:**

["NO or VERY MINOR ethics concerns only"]

**Final Justification:**

- The authors provide a comprehensive mathematical background, detailed derivations, and clear pseudocode, enabling full understanding and reproducibility of their approach.
-  Empirical Validation and Utility: The method is validated on both synthetic dynamical systems and real neural data,  providing mechanistic insight into complex dynamics.
- The concern about generalizing to non-PL scenarios is properly addressed by providing an extension to tanh.

**Limitations:**

Yes

**Quality:**

3

**Strengths And Weaknesses:**

**Strengths**
- The authors provide comprehensive mathematical background, detailed derivations, and clear pseudocode, enabling full understanding and reproducibility of their approach.
-  Empirical Validation and Utility: The method is validated on both synthetic dynamical systems and real neural data,  providing mechanistic insight into complex dynamics.

**Weakness**
- The approach is specifically designed for ReLU-based, piecewise-linear RNNs and may not generalize to RNNs with smooth or other types of nonlinear activations, limiting its broader applicability.

---

> ### Author Rebuttal · Authors · 2025-07-30
>
> We thank the referee for the supportive and constructive review!
>
> **Weaknesses/ questions**
>
> While it is true that our algorithm exploits the particular properties of piecewise linear (PL), ReLU-based, RNNs, we would like to point out that this nevertheless encompasses a broad  class of current SOTA dynamical systems reconstruction (DSR) models, such as almost-linear (AL) RNNs [1], shallow PLRNNs [2], dendritic PLRNNs [3], or recurrent switching linear DS [4,5], and that, more generally, PL models have a famous history in the mathematics of DS [6,7] and in engineering [8,9] for their tractability properties. We believe this is a huge advantage of using PL models in this field of DSR, for which our algorithm is intended.
>
> This being said, we looked into the possibility of extending our algorithm to RNNs with smooth activation functions such as tanh, and provide a modified algorithm for such cases below which rests on local linearizations. The suggested algorithm proceeds by locally linearizing the manifold from a set of points by PCA. A set of support points is then propagated forward (for unstable manifolds) or backward (for stable manifolds) via the RNN flow map (for which we can encourage invertibility via regularization). As long as these points remain well-approximated by the current locally linear manifold as defined by PCA, we continue propagating within the same linear patch. When the current linear model no longer provides an adequate approximation, as determined by a threshold on the reconstruction error (see below), a new locally linear manifold is constructed via PCA on these propagated points.
>
> While this procedure is no longer exact (as for PL models) and computationally more demanding (since we cannot determine the manifold in an entire linear subregion but only locally within a patch), it still retains some of the scalability and interpretability of the original approach, without requiring a PL form of the RNN. To determine when the locally linear approximation becomes inaccurate, we compute the PCA reconstruction error on the newly propagated points $S\_\\text{new} = \\{ s\_i\\}\_{i=1}\^N$ assuming the current local patch $Q\_k$ (see [10]). If $\\text{err}\_\\text{rec} > \\epsilon\_\\text{rec}$, we start a new manifold patch by fitting a new local PCA model. This test detects whether the manifold curvature significantly increases and the support points start deviating from the current locally linear structure.
>
> We tested this idea on a vanilla RNN with tanh activation trained on the Duffing system which has a curved stable manifold (cf. Fig. 4a), and observed that the reconstructed stable manifold of the saddle agreed very well with the model’s vector field (as in Fig. 4a). While unfortunately the current rebuttal policies don’t allow for including additional graphs, we will include these results in the revision.
>
> **Algorithm R1:** *Manifold Construction for Smooth RNNs*
>
> **Input:**
> &nbsp;&nbsp;&nbsp;&nbsp;$N_\{max\}$: Maximum number of iterations
> &nbsp;&nbsp;&nbsp;&nbsp;$P$: periodic point
> &nbsp;&nbsp;&nbsp;&nbsp;$σ$: Index indicating whether the stable or unstable manifold is to be computed
> &nbsp;&nbsp;&nbsp;&nbsp;$E^\sigma$: Stable/unstable eigenvectors
> &nbsp;&nbsp;&nbsp;&nbsp;$\lambda^\sigma$: Stable/unstable eigenvalues
> &nbsp;&nbsp;&nbsp;&nbsp;$\epsilon$: Threshold for deciding when to create a new patch
>
> **Output:**
> &nbsp;&nbsp;&nbsp;&nbsp;$\\{S^\sigma_k, Q^\sigma_k\\}^{k_{max}}_{k=0}$: Sampled points and local manifold models
>
> ---
>
> 1. $k \leftarrow 0$ ▷ Index of current manifold patch
> 2. $Q^\sigma_k \leftarrow \text{GetManifold}(P, E^\sigma, \lambda^\sigma)$ ▷ Initialize linear manifold
> 3. $S^\sigma_k \leftarrow \text{SamplePoints}(Q^\sigma_k)$ ▷ Sample initial points on manifold
> 4. **for** $n = 0 : N_{\text{max}}$ **do**
>    - $S^\sigma_{\text{new}} \leftarrow \text{Propagate}(S^\sigma_k, \sigma)$ ▷ Propagate forward/backward depending on $\sigma$
>    - $err_\{rec\} \leftarrow ReconstructionError(S^\sigma_\{new\}, Q^\sigma_k)$ ▷ PCA reconstruction error
>    - **if** $\text{err}_{\text{rec}} > \varepsilon$ **then**
>      - $k \leftarrow k + 1$
>      - $Q^\sigma_k \leftarrow \text{PCA}(S^\sigma_{\text{new}})$ ▷ New local manifold patch
>      - $S^\sigma_k \leftarrow \text{SamplePoints}(Q^\sigma_k)$ ▷ Resample within new patch
>    - **end if**
> 5. **end for**
> 6. **return** $\\{S^\sigma_k, Q^\sigma_k\\}^{k_{max}}_{k=0}$ ▷ Return patches and samples
>
> **References:**
>
> [1] Brenner et al. Almost‑Linear Recurrent Neural Networks Yield Highly Interpretable Symbolic Codes in Dynamical Systems Reconstruction. In NeurIPS 2024
>
> [2] Hess et al. In Proceedings of the 40th International Conference on Machine Learning, pages 13017–13049. PMLR, July 2023.
>
> [3] Brenner et al. Tractable Dendritic RNNs for Reconstructing Nonlinear Dynamical Systems. In Proceedings of the 39th International Conference on Machine Learning, pages 2292–2320. PMLR, June 2022.
>
> [4] Linderman et al. Bayesian Learning and Inference in Recurrent Switching Linear Dynamical Systems. In Proceedings of the 20th International Conference on Artificial Intelligence and Statistics, pages 914–922. PMLR, April 2017.
>
> [5] Alameda-Pineda et al. Variational Inference and Learning of Piecewise Linear Dynamical Systems. IEEE Transactions on Neural Networks and Learning Systems, 33(8):3753–3764, August 2022
>
> [6] Alligood et al. Chaos: An Introduction to Dynamical Systems. Textbooks in Mathematical Sciences. Springer, 1996.
>
> [7] Lind et al. An Introduction to Symbolic Dynamics and Coding. Cambridge University Press, Cambridge, 1995.
>
> [8] Carmona et al. On simplifying and classifying piecewise-linear systems. IEEE Transactions on Circuits and Systems I: Fundamental Theory and Applications, 49(5):609–620, 2002.
>
> [9] Sontag. Nonlinear regulation: The piecewise linear approach. IEEE Transactions on Automatic Control, 26(2):346–358, April 1981
>
> [10] Reiss and Wahl. Nonasymptotic upper bounds for the reconstruction error of PCA. The Annals of Statistics 48, no. 2 (2020): 1098-1123.

---

> > ### Comment · Reviewer_1awS · 2025-08-05
> >
> > I thank the authors for their detailed response. My concern has been addressed.

---

> ### Author Response · Authors · 2025-08-05
>
> We are happy we were able to address your concerns, and thank you once again for your feedback and great suggestion to extend our algorithm, making our results much more widely applicable now.

---

### Public Comment · ~Lukas_Eisenmann1 · 2025-10-30

We are disappointed that this paper was rejected although it received unequivocal support from all four referees, with all scores on acceptance (5,4,4,4). In their final justification, all referees state that all open issues had been resolved by us during the rebuttal. Hence, from the referees’ side there is no indication as to why this paper could have been rejected, or which aspects of it would require further improvement.

From the AC summary, this was not clear to us either. The only reason stated in the summary is “Nonetheless, from the perspective of an AC, I feel the paper lacks some high-level and deep understanding of the geometry/topology of the manifold of RNN dynamics, which may limit its generalizability and scalability in the long run.” We do not understand what is meant by this statement, i.e. what exactly this “high-level and deep understanding” is supposed to be the AC is missing, and why and how this would affect generalizability or scalability. In our paper, we had started from the most general mathematical definition of un-/stable manifolds of any fixed or cyclic point (Def. 1, p. 3), and then gave a mathematically precise prescription for how to determine these manifolds. We also discussed in detail how these objects are crucial for the system dynamics. We do not understand where and in what sense this could have been “higher level” or “deeper”, and why this constituted sufficient reason for rejection after a very successful rebuttal round.

---

### Decision · Program_Chairs · 2025-09-17

**Decision:**

Reject

**Comment:**

The paper proposes an algorithm for detecting stable and unstable manifolds of fixed and cyclic points in piecewise-linear RNNs with ReLU activation functions. The authors claim this is the first algorithm for detecting stable/unstable manifolds in ReLU-based RNNs. The paper then utilizes some dynamical system examples such as Lorenz-63 attractors and single-cell recordings to illustrate the proposed algorithm.
The reviewers questioned the extension of the algorithm to RNNs with other nonlinear activation functions and the regularization on invertibility. The authors’ rebuttal has successfully addressed most of these concerns, including providing the strategy in the case of tanh activation functions. All of these support the strength of the paper on the algorithmic level. Nonetheless, from the perspective of an AC, I feel the paper lacks some high-level and deep understanding of the geometry/topology of the manifold of RNN dynamics, which may limit its generalizability and scalability in the long run. So, after a long-time debate, I am regretful to decide not to accept this paper to fit the NeurIPS capacity. But it should not be treated as if this paper is not technically sound in the algorithmic level, and I hope the paper has a smooth resubmission.